# The impact of the Agulhas Current System on precipitation in southern Africa in regional climate simulations covering the recent past and future

Nele Tim[1], Eduardo Zorita[1], Birgit Hünicke[1], and Ioana Ivanciu[2]

[1]Helmholtz-Zentrum Hereon, Institute of Coastal Systems - Analysis and Modeling, Max-Planck-Strasse 1, 21502 Geesthacht, Germany
[2]GEOMAR Helmholtz Centre for Ocean Research Kiel, Kiel, Germany

**Correspondence:** Nele Tim (nele.tim@hereon.de)

**Abstract.** Southern African climate is strongly impacted by climate change. Precipitation is a key variable in this region as it is linked to agriculture and water supply. Simulations with a regional atmospheric model over the last decades and the 21st century display a decrease in the past precipitation over some coastal areas of South Africa and an increase over the rest of southern Africa. However, precipitation is projected to decrease over the whole southern part of the domain in the future. This
study shows that the Agulhas Current System, including the current and the leakage, which surrounds the continent in the east and south, impact this precipitation trend. A reduction in the strength of the Agulhas Current is linked to a reduction in precipitation along the southeast coast. The Agulhas leakage, the part of the Agulhas Current that leaves the system and flows into the South Atlantic, impacts winter precipitation in the southwest of the continent. A more intense Agulhas leakage is linked to a reduction in precipitation in this region.

## 1 Introduction

Precipitation is an important factor in the southern African climate. It defines the seasons and it directly impacts one of the principal sources of income, agriculture. Southern Africa is located between tropical and temperate systems. Hence, the latitudinal shift in the position of the Intertropical Convergence Zone and the westerlies south of the continent, as well as the location of the anticyclones over the South Atlantic and western Indian Ocean, impact rainfall over southern Africa (Reason, 2017). Most
of southern Africa receives most of the rainfall in austral summer (Reason, 2017). This Summer Rainfall Zone (SRZ) extends from the tropics to the Southern Hemisphere midlatitudes (Chevalier and Chase, 2016). Only the uttermost southwest of the continent receives most of its annual rainfall in the winter season (Chevalier and Chase, 2016). In this Winter Rainfall Zone (WRZ) rainfall is caused mainly by frontal systems moving with the westerlies (Reason, 2017).

Southern Africa is surrounded by the Indian and the South Atlantic Oceans. Along its east and south coasts flows the Agulhas
Current. This warm water current flows through the Madagascar Street southeastward along the coast, overshoots when reaching the southernmost tip of the continent, the Cape Agulhas, and turns back into the Indian Ocean in the Retroflection area southwest of the continent as the Agulhas Return Current (Beal et al., 2011). A small fraction of the Agulhas Current leaves the

Cape Basin and continues into the South Atlantic (Tim et al., 2018). This proportion is called Agulhas leakage. The Agulhas Current System impacts the precipitation in southern Africa (Imbol Nkwinkwa et al., 2021; Nkwinkwa Njouodo et al., 2018;

Cheng et al., 2018; Jury, 2015; Reason, 2001). Especially for the southeast coast, the air masses that reach the continent accumulate heat and moisture by travelling from the Indian Ocean over the warm Agulhas Current towards the continent, favouring precipitation in that region (Imbol Nkwinkwa et al., 2021). The sea surface temperatures (SST) of the Agulhas Current also impact South African precipitation indirectly via the El Niño-Southern Oscillation (ENSO) (Jury, 2015). The Benguela Upwelling System, one of the four Eastern Boundary Upwelling Systems, is located along the southwest coast. Its cold surface

waters impact the moisture and heat content of the air masses above, leading to foggy conditions along the shoreline and desert conditions further inland (Reason, 2017).

Due to its unique location between the trades and westerlies and the currents along its coast in the adjacent oceans, climatic changes in this region are linked to changes in these various drivers. Precipitation over the past decades displays long-term changes. Precipitation trends in the WRZ based on observations depend on the time period and on whether annual means or the

actual rainfall season is analysed. A drying is detected for the winter rainfall season over the recent past (1987-2016) (Roffe et al., 2021) whereas a wettening was found over longer periods (1960-2010) for the winter rainfall season alone and annual precipitation (MacKellar et al., 2014). Wolski et al. (2021) found negative trends of annual means for the long period 1900-2017 and the recent past 1981-2017, positive trends for the period 1933-2014, and mixed trends for the periods 1981-2014 and 1933-2017. This shows that trends over periods including the drought 2015-2017 are generally negative. Ndebele et al.

(2020) analysed the observed rainfall of one station in Cape Town and found again that trends depend on the considered period. Rainfall increased over the periods 1841-1900 and 1930-1970 but decreased when the long period from 1900 to 2016 is analysed. Gridded observational data indicate mostly an increase in the winter rainfall season for several time periods (Onyutha, 2018). Precipitation in the SRZ in austral summer (December-January-February, DJF) has either increased or no trend has been detected. This happens in both types of datasets, station observations and gridded fields (Onyutha, 2018; Kruger and Nxumalo,

2017; MacKellar et al., 2014).

The prevailing wind systems, the trade winds and westerlies, have also shifted in the past and are projected to further shift poleward (Tim et al., 2019). This may not only cause changes in precipitation directly but probably also indirectly via changes in the Agulhas Current System. The Agulhas Current has weakened (Schwarzkopf et al., 2019) and shifted poleward (Yang et al., 2016), the Agulhas leakage has intensified (Tim et al., 2018) and both are projected to continue their past trends in the

21st century (Ivanciu et al., 2022a). As southern Africa is prone to changes via anthropogenic climate change due to changes in the wind systems, the ocean currents, and air and ocean temperatures, we investigate here the current state of precipitation, the past and future precipitation trends, and the impact of the Agulhas Current System on precipitation.

For these purposes, we analyse several simulations with a high-resolution (16 km horizontally) regional atmospheric model.

This high resolution allows us to more faithfully represent the processes leading to precipitation than the coarser resolution of global climate models (typically 100 km) and of regional climate models (about 50 km) that participate in the Coupled Model Intercomparison Project (CMIP6) used by the IPCC in their climate assessment reports (Lee et al., 2021). Our model set-up

includes, in addition to the regional atmospheric model, a global coupled climate model that provides the boundary conditions for the regional simulation. The resolution of the ocean component is also refined in the region surrounding southern Africa in comparison with usual global simulations so that it allows for a better representation of the Agulhas Current and the Agulhas leakage (Schwarzkopf et al., 2019). Therefore, this model set-up allows to estimate the strength of the Agulhas Current and the Agulhas leakage and to statistically analyse the connections between the Agulhas Current System and the SST, sea level pressure (SLP) and precipitation in this region.

The results of the Coupled Model Intercomparison Project (CMIP6) as summarised in the Sixth Assessment Report by the IPCC indicate that precipitation in this region will tend to decrease in the future, especially in the WRZ (see Figure 4.32 in the full report by Working Group 1 (Lee et al., 2021)). The projected reduction in precipitation is robust across the ensemble of climate models and becomes more intense with increasing concentrations of greenhouse gases in the future. In fact, southern Africa and the Mediterranean countries are the two land regions that are projected to suffer from the strongest precipitation reductions in the future. Figure 4.4 in the Sixth Assessment Report (Lee et al., 2021) clearly shows that the southern hemisphere land areas where annual precipitation is projected to decrease are located in the subtropical western continental regions: western Australia, Chile, and western southern Africa. In other land areas in the southern hemisphere, the projected precipitation changes are small. A common large-scale mechanism related to the increased radiative forcing and independent of changes in the Agulhas Current System is likely behind this large-scale spatial pattern of precipitation reduction. Previous studies have identified a diminished zonal moisture advection from the oceans located further west (Seager et al., 2019). However, in addition to that large-scale mechanism, the regional ocean circulation patterns may also modulate the projected precipitation trends, and these regional mechanisms should also be considered and analysed with higher-resolution models. Since the resolution of the global climate models, including the ocean component, might not be totally adequate to resolve important processes for southern African precipitation (Munday and Washington, 2018; Jury, 2012), as indicated above, a study with higher resolution models is important to confirm this future reduction in precipitation and to assess the contribution of regional ocean processes.

As far as we know, up to now, there has been only one study by Cheng et al. (2018) investigating the impact of the strength of the Agulhas leakage on the regional climate in the past decades. This study was mostly focused on the impact of the Agulhas leakage on the interannual regional patterns of SST and precipitation over the ocean and adjacent land masses. Cheng et al. (2018) also analysed a coupled ocean-atmosphere simulation with a high-resolution eddy-resolving ocean component. They found that a stronger leakage causes warmer surface water temperatures in the retroflection zone and colder water temperatures in the Indian Ocean further east. The impact of these temperature patterns is seen in stronger convective precipitation over the warmer ocean (and weaker over colder ocean regions). On the adjacent land, a stronger leakage causes weaker convective precipitation in the SRZ.

The present study has a broader scope. We also consider herein the future impact of the Agulhas Current, not only at interannual (or multi-annual) time scales in the historical period but also at the longer-term centennial scales under increasing concentrations of greenhouse gases.

## 2 Data and methods

The data used for this analysis encompasses regional atmospheric model simulations, gridded observational data sets, atmospheric reanalysis data, and global coupled ocean-atmosphere simulations. As observational data sets, we use the Global Precipitation Climatology Project (GPCP, version v01r03, Adler et al. (2003)), the Global Precipitation Climatology Centre (GPCC, version v.2020, Schneider et al. (2016)), precipitation data from the Climate Research Unit (CRU, version ts4.05, Harris et al. (2020)), and the Hadley Centre Global Sea Ice and Sea Surface Temperature (HadISST1, Rayner et al. (2003)). GPCP is a combination of observations with satellite measurements of precipitation in 1-degree horizontal resolution and covers the period from October 1996 to present. GPCC is based on in situ rain-gauge data, has 0.25-degree horizontal resolution, is available from 1981 onwards, and is the observational part of GPCP. CRU is also based on station data, has 0.5 degree horizontal resolution, and covers the period 1901 to today. HadISST has a horizontal resolution of 1 degree and is available from 1871 to present.

The Japanese reanalysis data set JRA-55 (Kobayashi et al., 2015) serves for the model validation as it is the driving data of the regional model CCLM (COSMO model in CLimate Mode, $https://www.clm-community.eu/$), as described in the following section. It starts in 1958 with a spatial horizontal resolution of 0.56 degree. Additionally, we use the reanalysis data set ERA5 (Hersbach et al., 2020) for validation of the precipitation trend. ERA5 data starts in 1979 with a horizontal resolution of 31 km.

In the context of the CASISAC project (Changes in the Agulhas System and its Impact on southern African Coasts, funded by the German Federal Ministry of Education and Research BMBF), we performed three regional atmospheric simulations over southern Africa (covering the region 10°W - 55°E, 0°S - 55°S, Fig. 1) with the CCLM model. We set up one hindcast simulation where JRA-55 (Kobayashi et al., 2015) is driving the simulation at the initial and lateral boundaries. This simulation spans the period from January 1958 until April 2019. The other two CCLM simulations are driven by the coupled climate model FOCI (Flexible Ocean and Climate Infrastructure, Matthes et al. (2020)) with interactive ozone chemistry, and high, mesoscale-resolving, ocean resolution around South Africa, run by our project partners at the research centre GEOMAR (Germany). These encompass a historical simulation (1951-2013) and a scenario simulation with increasing greenhouse gas concentrations (2014-2099, SSP5-8.5 (Lee et al., 2021)). These CCLM simulations have a horizontal resolution of about 16 km and an hourly output. We used spectral nudging (von Storch et al., 2000) for all three simulations. Thus, the large-scale pattern of the driving data sets is nudged over the whole model domain. This method is beneficial as small-scale features are developing freely and large-scale features are kept tight to the driving data not only at the boundaries. Time series of the Agulhas Current and Agulhas leakage transport were obtained from the coupled climate model FOCI using Lagrangian particle tracking technique, as described by Ivanciu et al. (2022a).

To analyse the changes in precipitation over the past decades and the 21st century, we calculated the linear trend. For the statistical significance of these trends, a significance level of p=0.05 was adopted using the method f-test, a test for the null hypothesis that the variances of two normal populations are the same.

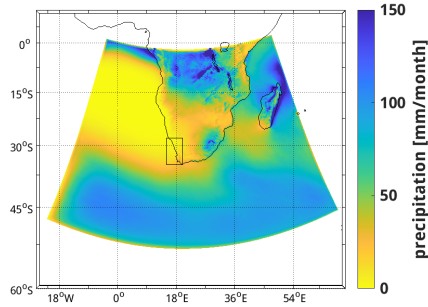

**Figure 1.** The CCLM domain of this study is shown as the climatology of rainfall over the hindcast period 01/1958-04/2019.

To estimate the impact of the Agulhas Current and Agulhas leakage on SST and precipitation in the CCLM simulations, a simple linear regression model was assumed:

$$Prec(t) = Prec_{mean} + a \times AL(t) + b(t) \tag{1}$$

where $Prec(t)$ is the precipitation in a model grid-cell over a 5-year period starting in year $t$, $Prec_{mean}$ is the climatological mean precipitation in that model grid-cell, $AL(t)$ is the transport of the Agulhas leakage in the 5-year period starting in year $t$. The parameter $a$ is the regression slope and $b(t)$ is the part of the precipitation in the 5-year period that is not related to the Agulhas leakage. The regression parameters (for each grid cell) can be estimated from the simulated values of the precipitation and the Agulhas leakage by ordinary least-squares regression. The portion of the variability in precipitation that can then be linearly attributed to the Agulhas leakage, $Prec_{AL}$, is

$$Prec_{AL}(t) = a \times AL(t) \tag{2}$$

and its long-term trend is simply the linear trend of $Prec_{AL}$. The same equations apply for the SST by replacing $Prec$ with $SST$ and replacing the Agulhas leakage (AL) with the Agulhas Current. For the Agulhas Current, no 5-year period was used but the strength of each individual year.

## 3 Mean precipitation and validation of CCLM simulations

In this section, we validate the precipitation simulated by the hindcast CCLM simulation over our model domain with the observational data sets and the JRA-55 reanalysis data set and provide a general overview of the rainfall over southern Africa. The climatology of simulated precipitation represents well the different climatic regions in southern Africa (Fig. 1). Regions of higher rainfall amounts are located in the tropics, over Madagascar, and along the southeast coast of South Africa. Dryer regions are the Namib and Kalahari deserts in the WRZ. South Africa can be divided into 8 rainfall zones: the North-Western Cape and the South-Western Cape constitute the WRZ, the South Coast, which has similar rainfall amounts during all months

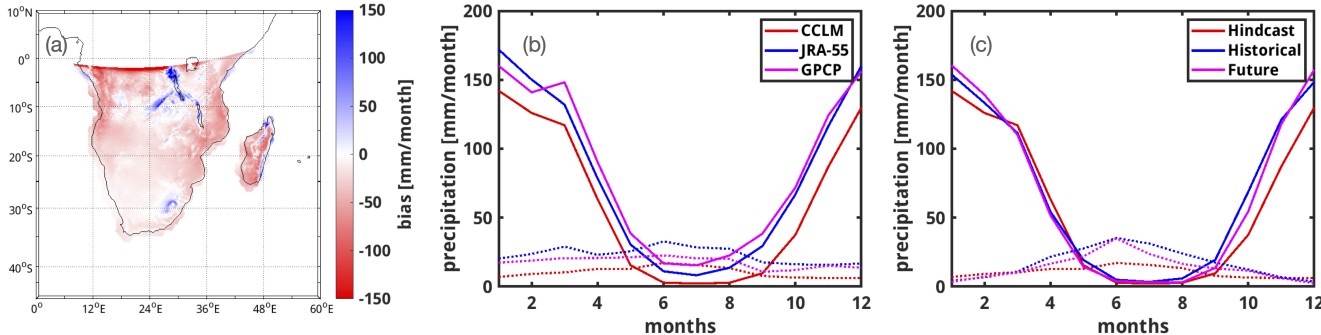

**Figure 2.** Model validation: (a) model rainfall bias in [mm/month] given as the difference between modelled and observed rainfall from CCLM and GPCP, respectively, for the overlapping period 1997-2018 (CCLM-GPCP). Annual cycle of rainfall over the Summer Rainfall Zone (SRZ) and the Winter Rainfall Zone (WRZ) for (b) the hindcast CCLM simulation (red), the reanalysis data set JRA-55 (blue), and the GPCP observational data set (pink) and for (c) the three CCLM simulations hindcast (red), historical (blue), and future (pink). Solid lines represent the rainfall for the SRZ, dashed lines represent the WRZ.

of the year (all-year rainfall zone (YRZ)) and the SRZs Southern Interior, the Western Interior, the Central Interior, KwaZulu-Nata and the North-Eastern Interior (Rouault and Richard, 2003).

Comparing CCLM to GPCP shows that CCLM generally underestimates precipitation amounts (e.g. over Madagascar) except

150    for the Drakensberg region in eastern South Africa and the large lakes in the northeast of our domain (Fig. 2a). Differences over Madagascar, the Congo basin, and the coasts north of 20°S are probably due to a misrepresentation of the African monsoon system. Comparing CCLM to other observational data sets that cover a longer period but are available at monthly resolution, like GPCC and CRU, provides a very similar picture (see Fig. A1). The bias of CCLM given as the percentage of the mean rainfall (relative bias) exhibits a similar magnitude when computed with respect to all three observational data sets (GPCP,

155    GPCC, and CRU) (Fig. A2). Considering the relative bias reveals that deviations of simulated precipitation from observed ones are noticeable in the whole domain, also in our focus region, the southern part of the model domain, and along the coasts, along which the Agulhas Current flows. This general underestimation of rainfall amounts in CCLM can also be seen in the annual cycle (Fig. 2b). CCLM represents well the shape of the annual cycle of both rainfall regions SRZ and WRZ. The WRZ region in this study covers the domain 15°E - 20°E and 28°S - 35°S. Rainfall peaks in December and January in the SRZ and in June

160    in the WRZ. However, comparing the hindcast simulation to its driving data set JRA-55 and to the observational data set GPCP, shows that CCLM simulates generally lower rainfall amounts than the other two data sets. JRA-55 also underestimates slightly the observations and these too low rainfall amounts remain and intensify in CCLM. Both the WRZ and the SRZ in this study include parts of the YRZ. This relatively small transition zone between the WRZ and SRZ is not analysed separately here. This may contribute to the relatively flat annual cycle of the WRZ.

165    Thus, the simulated precipitation of CCLM represents well the rainfall zones of southern Africa covered by the model domain. It slightly underestimates the precipitation amount compared to the driving reanalysis data set and observations in most of the domain.

## 4 Trends in precipitation

In this section, we compare past precipitation trends in CCLM to the observational data sets CRU, GPCC, and GPCP and to the reanalysis data sets JRA-55 and ERA5 and we investigate the future precipitation changes under the SSP5-8.5 emission scenario. We aim to answer the following questions: How has precipitation changed over the past over the WRZ and SRZ? Are there regional differences? And how is the rainfall projected to change until the end of the 21st century?

Precipitation trends are shown for austral summer in figure 3 and for austral winter in figure 4.

Regarding austral summer, changes in precipitation over southern Africa look quite similar for CRU and GPCC but different from GPCP and the reanalysis data sets JRA-55 and ERA5. For the southern part of the domain, CRU and GPCC show weak trends of both wettening and drying (Fig. 3a, b). GPCP shows stronger trends and rather a reduction in precipitation, especially over Madagascar and the east coast (Fig. 3c), whereas JRA-55 and ERA5 indicate an increase in precipitation over Madagascar and western southern Africa (Fig. 3d, e). The results from CRU, GPCC, and JRA-55 are plotted here for the same period 1958-2019. The deviations of GPCP might be due to the shorter period covered by this data set (1997-2018). The reanalysis data set ERA5 starts in 1979, later than JRA-55. This might lead to the weaker wettening trends in the southern part of the domain.

CCLM generally simulates trends of reduced strength compared to its driving data set JRA-55. The hindcast simulation shows an attenuated but generally similar pattern compared to JRA-55 (Fig. 3f). The FOCI-driven simulation over the historical period (Fig. 3g) provides a somewhat similar precipitation trend for the southern part of the domain but with even smaller regions of intensification than the hindcast and JRA-55, similar to CRU and GPCC.

Overall, since 1958, the hindcast and reanalysis data sets show stronger trends with drying for parts of the southeast coast and west coast and wettening for the south coast (the YRZ), and central southern Africa for austral summer. CRU indicates wettening in the east (including the YRZ part of the SRZ) and drying in the west. The historical simulation agrees well with GPCC showing weak trends of both drying and wettening.

Trends in the future climate simulation (Fig. 3h) are weak too. Precipitation is projected to decrease in the southern part of the domain, also along the coast. An intensification is simulated for Madagascar and the east coast of the African continent along the Mozambique channel.

For the austral winter (Fig. 4), GPCC, GPCP, JRA-55, ERA5, and the hindcast simulation show a wettening in the south of the WRZ and drying in the north. Here the hindcast simulation (Fig. 4f) shows a stronger trend than the other data sets. CRU, the historical and future simulations (Fig. 4a, g, and h) show drying in the whole WRZ.

Thus, precipitation in the SRZ and WRZ have mainly increased in the past in the respective season and decreased in some coastal areas, while for the future precipitation, trends are negative for South Africa for both regions and seasons. This projection agrees with the IPCC assessment of future precipitation based on the CMIP6 model ensemble.

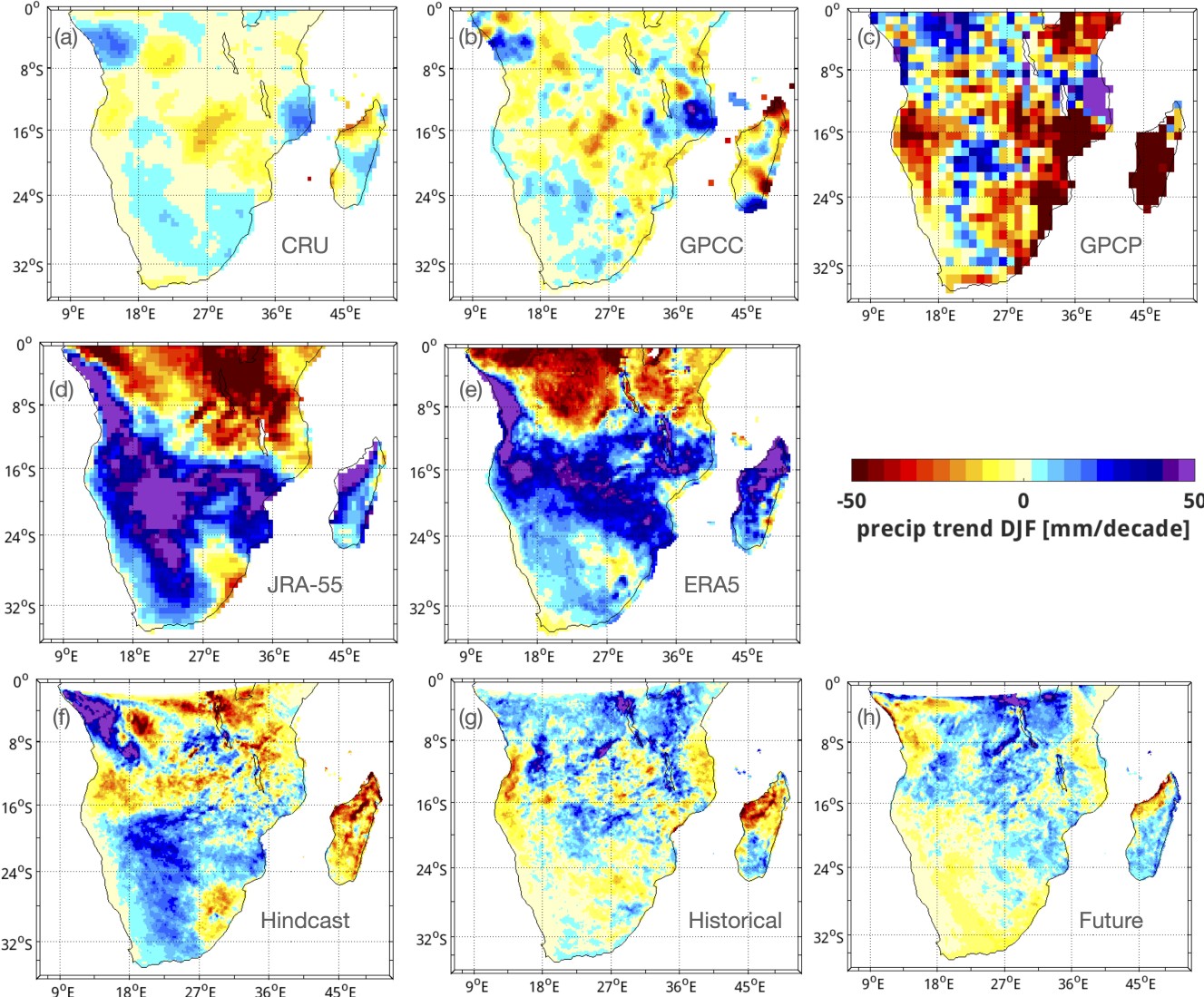

**Figure 3.** Precipitation trends for austral summer (December-February) for the Summer Rainfall Zone for the observational data sets (a) CRU (1958-2019), (b) GPCC (1958-2019), and (c) GPCP (1997-2018), for the reanalysis data sets (d) JRA-55 (1958-2019) and (e) ERA5 (1979-2020), (f) the hindcast simulation of CCLM (1958-2019), (g) the historical simulation of CCLM (1951-2013), and (h) the scenario simulation of CCLM (2014-2099). Trends are in mm/decade. Year dates are of January and February of the seasons.

The agreement with the IPCC projections is reflected in the simulated trends of the large-scale atmospheric circulation. The SLP patterns of the 1950s (Fig. 5b, 5c) compared to the ones of the 2090s (Fig. 5d, 5e) show that changes in austral summer of the subtropical highs in the South Atlantic and the western Indian Ocean are small, decreasing slightly close to the South African coast. The continental heat low over the Kalahari has intensified. SLP is lower over the Agulhas Current which may

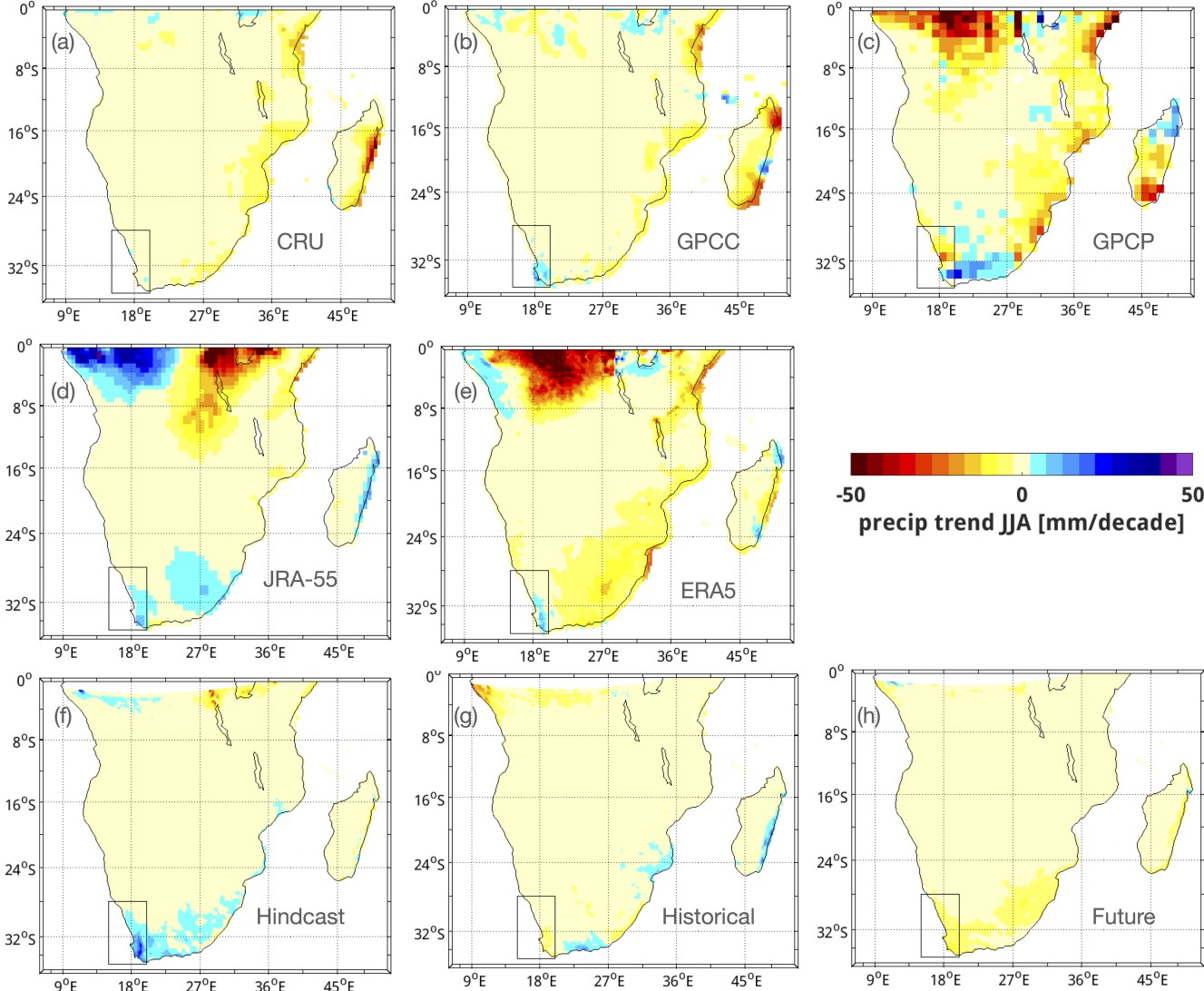

**Figure 4.** Precipitation trends of the winter rainfall season (June-August) for the Winter Rainfall Zone (marked by the box) for the observational data sets (a) CRU (1958-2019), (b) GPCC (1958-2019), and (c) GPCP (1997-2018), for the reanalysis data sets (d) JRA-55 (1958-2019) and (e) ERA5 (1979-2020), (f) the hindcast simulation of CCLM (1958-2019), (g) the historical simulation of CCLM (1951-2013), and (h) the scenario simulation of CCLM (2014-2099). Trends are in mm/decade.

hinder the inflow of moist warm air from the Indian Ocean and the development of tropical temperate troughs, responsible for summer rainfall. This may be linked to the decrease in precipitation over the southeast coast in austral summer. In austral winter, the SLP has increased over the oceans and the continent. Both subtropical highs have not only intensified but also shifted poleward. This, together with the poleward shift of the westerlies (Fig. 5a), leads to a more southward position of the frontal systems causing drying in the WRZ as we have seen in the precipitation trend analysis.


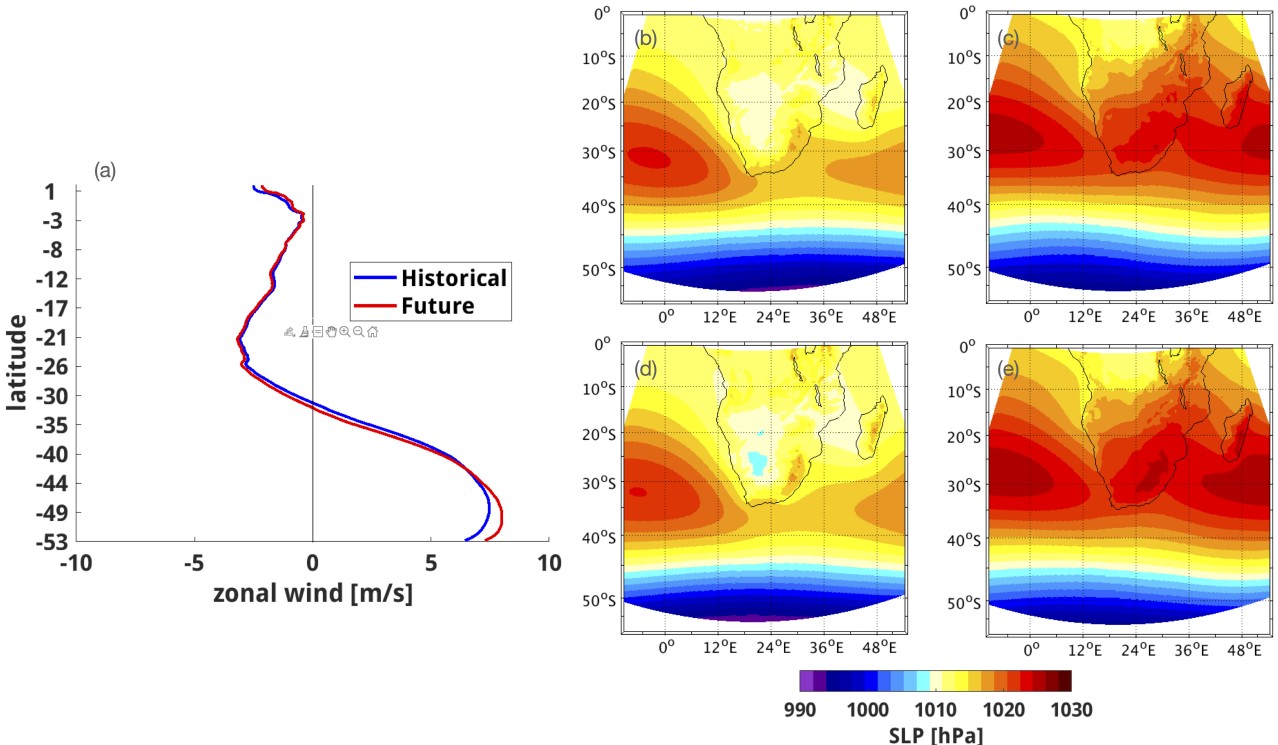

**Figure 5.** (a) The position and strength of the surface westerlies in the historical and scenario simulation as latitudinal plots (meridionally averaged over the model domain) of zonal wind strength. The SLP pattern for both rainfall seasons, (b) DJF, (c) JJA of the first 10 years of the historical simulation (1951-1960), (d) DJF and (e) JJA of the last 10 years of scenario simulation (2090-2099), in hPa.

There are two additional considerations. One is that the relatively weak trend in the future projected with the CCLM model might be an underestimation as this model produces weaker precipitation trends than observations and reanalysis data sets in the historical simulation. The second is that the historical and projected trends do not have the same sign in some regions, and thus the mechanism behind the observed historical trends may be partly unrelated to the increase of greenhouse gases.


## 5    The Agulhas Current System as a driver of precipitation

As we have seen in the previous section, precipitation trends are different on the coasts of southern Africa than inland. The nearby warm Agulhas Current System and its changes might influence these spatial patterns of the trend. Therefore, in this section, we investigate the relationship between the Agulhas Current System and the trends in precipitation over southern

Africa.

The Agulhas leakage exhibits a significant positive trend and the Agulhas Current exhibits a significant negative trend in both

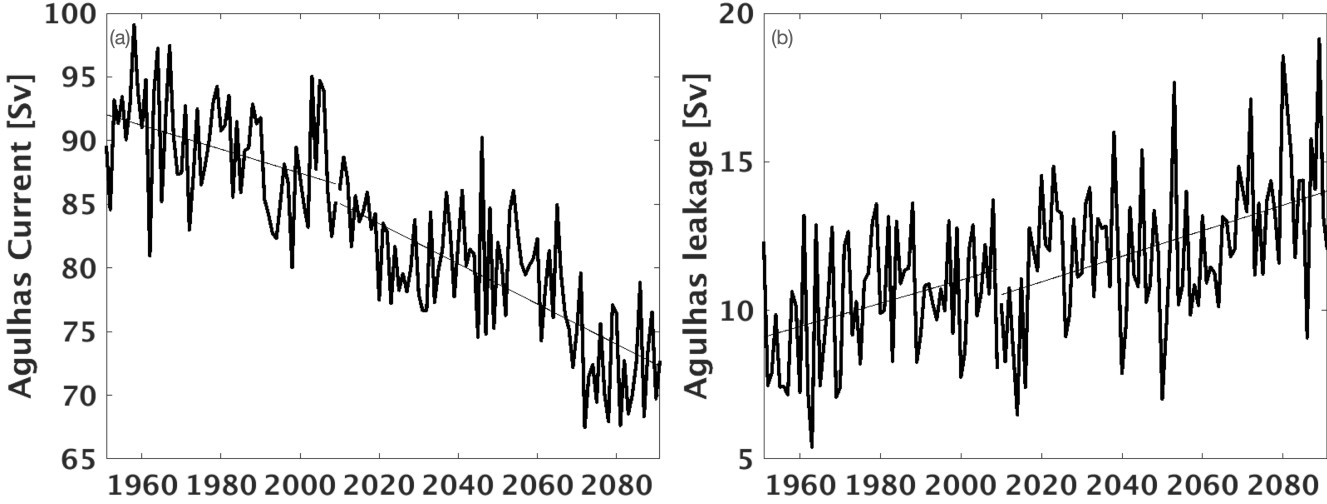

**Figure 6.** Time series and trend of the volume transport of the (a) Agulhas Current and (b) of the Agulhas leakage in the FOCI simulation, in Sverdrup [Sv].

FOCI simulations, the historical simulation covering the past and the scenario simulation covering the 21st century (Fig. 6). The trend in the Agulhas leakage has been further proven by using a proxy based on the SST of the CCLM simulation: The SST difference between the southwest Indian Ocean (36-40°S, 25-35°E) and the southwest Atlantic (34-40°S,10-20°E) (not

shown here). The SST difference was shown to be a good indicator for the Agulhas leakage intensity (Biastoch et al., 2015). Trends are significant for the hindcast simulation and for the past and future FOCI-driven simulations. Thus, the leakage has intensified whereas the Agulhas Current has reduced in strength and these trends are predicted to continue during the twenty-first century.

To detect the impact of the Agulhas Current System and its changes in strength on the precipitation over South Africa and the

neighbouring SST, we applied a linear regression model (described in section 2).

### 5.1 Attributing SST and precipitation to the strength of the Agulhas Current and Agulhas leakage

In this section, we look specifically at the impact of the strength of the Agulhas Current and Agulhas leakage on SST around southern Africa and precipitation over southern Africa. The Agulhas Current is expected to impact the precipitation of coastal southern Africa due to its closeness to the shore. The Agulhas Current is a region of high moisture and heat fluxes into the

atmosphere (Lee-Thorp et al., 1999). And since the Agulhas leakage determines the volume of warm and saline water masses from the Indian Ocean flowing into the South Atlantic, it can impact the SST, the atmospheric circulation and precipitation over southern Africa. The transport of the Agulhas Current and leakage is calculated from FOCI, the driving data set of the historical and scenario simulation of CCLM. The high resolution of FOCI around southern Africa enables the simulations of the mesoscale features which are important for the Agulhas Current System. The purpose of analysing both periods is to

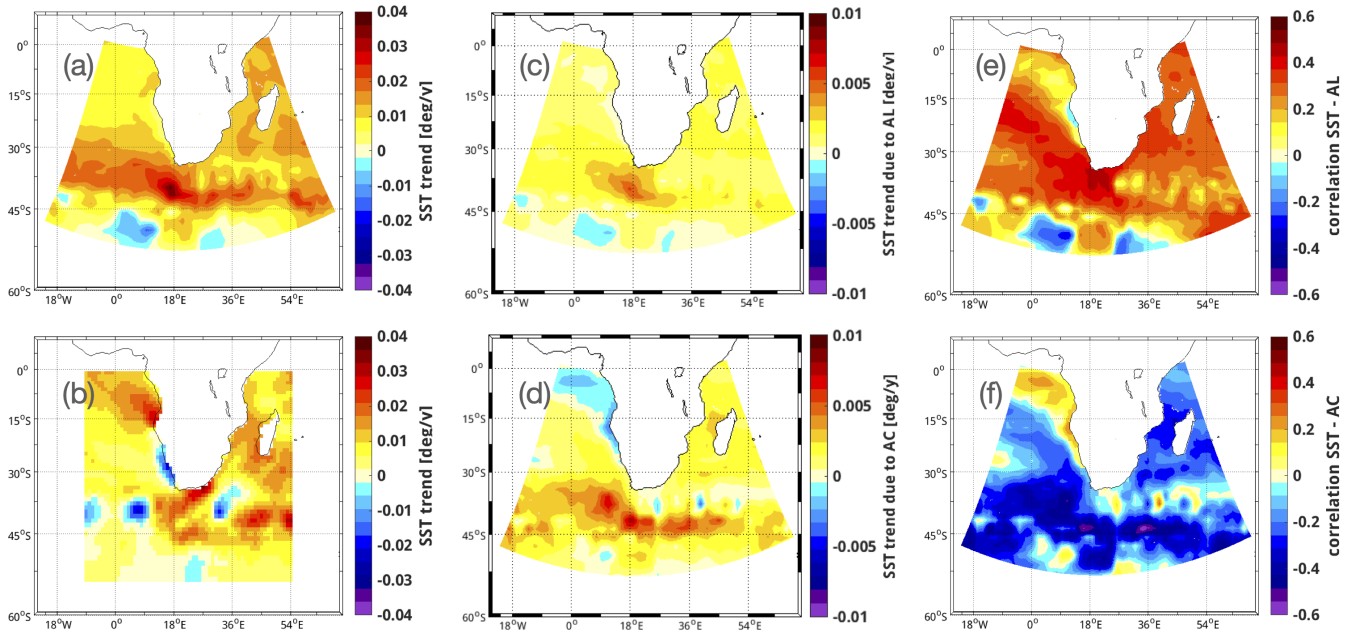

**Figure 7.** Impact of the Agulhas leakage (AL) and Agulhas Current (AC) on SST in the historical simulation. (a) The trend in SST of the historical simulation, (b) the trend in SST of the observational data set HadISST1, (c) the portion of the SST trend attributed to the Agulhas leakage, (d) the portion of the SST trend attributed to the Agulhas Current, (e) the correlation pattern of SST and the strength of the Agulhas leakage, and (f) the correlation pattern of SST and the strength of the Agulhas Current.

identify the impact of the Agulhas Current System on regional climate trends in the current climate on the one hand, and to estimate the contribution of changes in the Agulhas Current System on future climate change on the other hand.

The Agulhas Current is defined as the volume transport at a transect at 32°S. The Agulhas leakage is defined as the amount of water originating in the Agulhas Current at 32°S and crossing the Good Hope Line (Ansorge et al., 2005) within a 5 year window, thus leaving the Cape Basin and entering the South Atlantic (as described in Tim et al. (2018)).


Figure 7 shows the impact of the Agulhas Current and Agulhas leakage on the SST in the historical simulation. The SST trend shows warming, especially in the Retroflection area of around 0.03°C per year (Fig. 7a). Compared to the observed SST trend (Fig. 7b), the simulated trend does not show a cooling in the Benguela Upwelling System either in the west of the retroflection or between the Agulhas Current and the Return Current south of it. Nevertheless, both data sets, HadISST1 and CCLM, show

a warming of the Agulhas Current and in the area of retroflection. The Agulhas leakage and SST exhibit a positive correlation of 0.4 in the Agulhas Retroflection region, southwest of it and in the corridor where Agulhas rings transport Indian Ocean water into the South Atlantic (Fig. 7e). This reflects the transport of warm Indian Ocean water by the Agulhas Current and the farther pathway of this warm water into the South Atlantic due to the Agulhas leakage (Gordon, 1986). The increase in

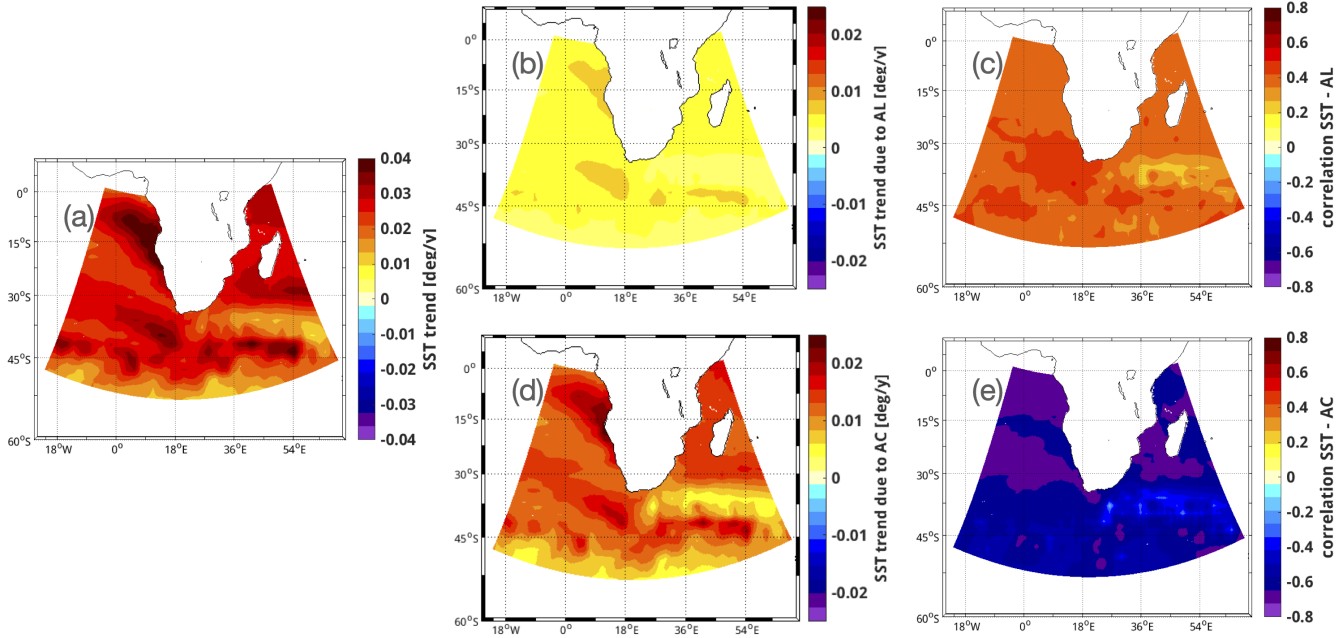

**Figure 8.** Impact of the Agulhas leakage (AL) and Agulhas Current (AC) on SST in the scenario simulation. (a) The trend in SST, (b) the portion of the SST trend attributed to the Agulhas leakage, (c) the correlation pattern of SST and the strength of the Agulhas leakage, (d) the portion of the SST trend attributed to the Agulhas Current, and (e) the correlation pattern of SST and the strength of the Agulhas Current.

Agulhas leakage over the last decades (Schwarzkopf et al., 2019; Tim et al., 2018; Durgadoo et al., 2013; Biastoch et al., 2009)
and the warming of the Agulhas Current System (Rouault et al., 2009, 2010) lead to a positive SST trend southwest of the Cape region. About 1/6 of this warming can be attributed to the Agulhas leakage (Fig. 7c).

The Agulhas Current also contributes to about 1/6 of the SST trend (Fig. 7d), in the Retroflection area and the region of the Agulhas Return Current. Correlations are mainly negative, indicating warming when the strength of the Agulhas Current is reduced (Fig. 7f). This opposite relationship between the strength of the Agulhas Current and the Agulhas leakage has been
previously found by van Sebille et al. (2009).

In the scenario simulation, the SST trend is similar to the trend in the past with a maximum of 0.04°C per year (Fig. 8a). The Agulhas leakage contributes again around 1/6 of this SST trend southwest of the Cape region, in the Agulhas Retroflection, and at the coast of northern Namibia and Angola (North Benguela Upwelling region) (Fig. 8b). The impact of the Agulhas leakage on the North Benguela Upwelling region is known to exhibit a lag of several years (Tim et al., 2018). The strong positive trends
in the Cape region and North Benguela Upwelling region are interlinked in another way. The poleward shift and intensification of westerlies and trades impact both oceanic regions: an intensification of the Agulhas leakage causes warming in that area (Biastoch and Böning, 2013) and a poleward shift in the upwelling region leads to reduced upwelling and consequently warmer temperatures off the coasts of Angola and northern Namibia (Rykaczewski et al., 2015).

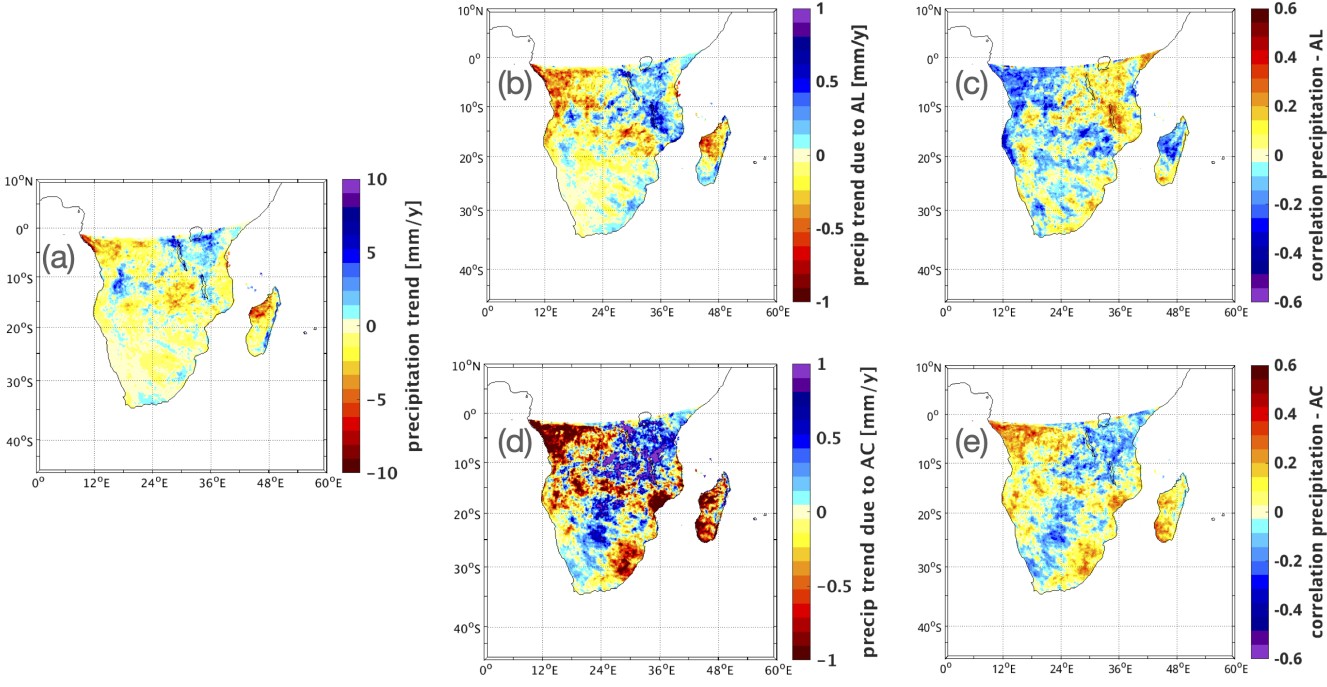

**Figure 9.** Impact of the Agulhas leakage (AL) and Agulhas Current (AC) on precipitation over southern Africa in the historical simulation. (a) The trend in precipitation, (b) the portion of the precipitation trend attributed to the Agulhas leakage, (c) the correlation pattern of precipitation and the strength of the Agulhas leakage, (d) the portion of the precipitation trend attributed to the Agulhas Current, and (e) the correlation pattern of precipitation and the strength of the Agulhas Current. Colour bars of the correlation patterns (c, e) are reversed compared to the colour bars of the other subplots.

In the scenario simulation, as for the historical period, the Agulhas leakage and SST are positively correlated with up to 0.5 in the Southeast Atlantic Ocean (Fig. 8c). Again as in the historical simulation, SST rise as the leakage is projected to increase under global warming (Ivanciu et al., 2022a; Biastoch and Böning, 2013).

The Agulhas Current also contributes to the warming in the Retroflection area (Fig. 8d) and correlations show, as for the historical period, that SST rise when the strength of the Agulhas Current is reduced (Fig. 8e). Nevertheless, the link between the strong SST trend in the Cape basin and the Agulhas Current occurs possibly via the Agulhas leakage. Also, the SST trends in the North Benguela Upwelling System are probably not directly impacted by the Agulhas Current, but rather climate change impacts both the Agulhas Current and the SST.

Thus, the Agulhas Current and the Agulhas leakage contribute to the warming trend of SST adjacent to southern Africa. The ocean surface has warmed and this trend is projected to continue. The weakening of the Agulhas Current and the intensification of the Agulhas leakage contribute to this warming for both past and future periods.

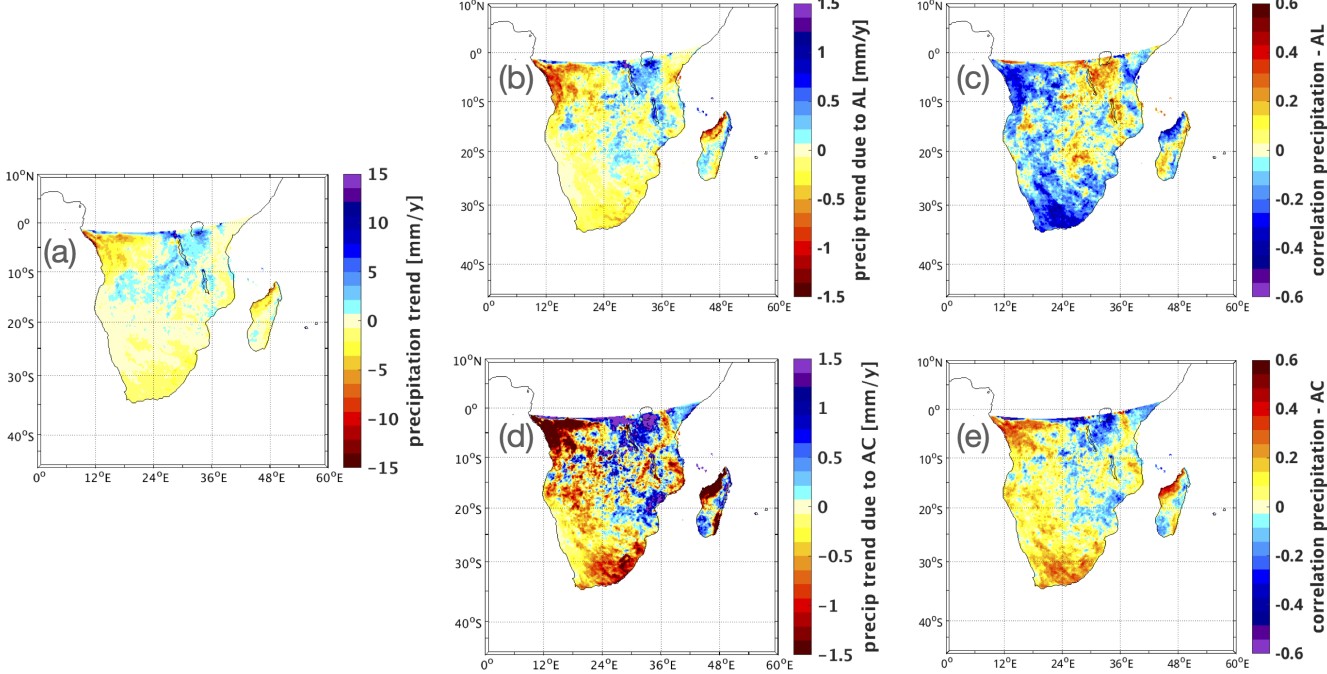

**Figure 10.** Impact of the Agulhas leakage (AL) and Agulhas Current (AC) on precipitation over southern Africa in the scenario simulation. (a) The trend in precipitation, (b) the portion of the precipitation trend attributed to the Agulhas leakage, (c) the correlation pattern of precipitation and the strength of the Agulhas leakage, (d) the portion of the precipitation trend attributed to the Agulhas Current, and (e) the correlation pattern of precipitation and the strength of the Agulhas Current. Colour bars of the correlation patterns (c, e) are reversed compared to the colour bars of the other subplots.

The impact of the Agulhas Current and Agulhas leakage on the precipitation over southern Africa in the past decades is shown in figure 9. Precipitation shows a weak negative trend in most of southern Africa, with some precipitation intensification along the southeast coast (Fig. 9a), and around 1/10 of this trend is due to the Agulhas leakage (Fig. 9b). Agulhas leakage and precipitation are positively correlated along the southeast coast of South Africa (Fig. 9c). The Agulhas leakage impacts precipitation
via the SST: SST in the Retroflection are positively correlated with precipitation at the southeast coast of South Africa (Cheng et al., 2018) and SST west of the Retroflection with precipitation in the SRZ (Walker, 1990). Correlations with precipitation in the WRZ around the Cape of Good Hope are negative (Fig. 9c).

The Agulhas Current causes a precipitation trend of opposite sign along the South African coast (Fig. 9d). Correlations are positive for coastal South Africa (Fig. 9e). Thus, a strong Agulhas Current is linked to enhanced rainfall, but as the Agulhas
Current has weakened, precipitation has decreased.

Regarding changes in precipitation in the future scenario, the Agulhas leakage and precipitation are negatively correlated over South Africa (Fig. 10c). Thus, as the leakage will intensify (Ivanciu et al., 2022a) it contributes to the reduction of precipitation along the whole coast and the southern inland in the future. In the past, a decrease in precipitation due to increased leakage only

occurred in the Cape region. This can be seen in figure 10a, which shows a projected weakening in precipitation in southern
Africa. Again, around 1/10 of this trend is due to the Agulhas leakage (Fig. 10b).

The analysis of the impact of the Agulhas Current on future precipitation indicates, like for the historical period, that the reduction of precipitation is partially caused by the decrease in the strength of the Agulhas Current (Fig. 10d, 10e).

The change in the dependency of precipitation on Agulhas leakage strength from the past to the future requires further investigation. A possible explanation could be that only the trend in precipitation in the WRZ is directly linked to the strength of the Agulhas leakage, whereas precipitation at the southeast coast is rather linked to the Agulhas Current. The signal of the weakening Agulhas Current may dominate the signal of the strengthening Agulhas leakage for the precipitation in the South African SRZ and especially for precipitation over the southeast coast of South Africa. An analysis of the impact of the Agulhas Current System on the winter rainfall (shown in Appendix B) indicates that the decline in austral winter rainfall in the past is driven by the Agulhas leakage and in the future by both Agulhas leakage and Agulhas Current. Another possible explanation might be that the Agulhas Current is only weakening at its current-day location (here measured at 32°S) and is shifting southward, away from the coast. This has been stated by Yang et al. (2016). Further, the precipitation at the southeast coast is linked to an Agulhas Current core located close to the coast (Jury et al., 1993). The strength of the Agulhas Current is linked to the strength of the trade winds (Loveday et al., 2014) and the strength of the Agulhas leakage is linked to the position and strength of the westerlies (Durgadoo et al., 2013; Beal et al., 2011). The westerlies have intensified and shifted poleward from the historical simulation to the scenario simulation, as already shown in figure 5a.

Thus, a decrease in precipitation took place over some parts of southern Africa in the past and is projected to do so for a large part of southern Africa in the future. This is linked to changes in both oceanic and atmospheric circulation. We have shown that the weakening of the Agulhas Current, both in the past and in the future, is associated with a reduction in precipitation, especially along the southeast coast, whereas the increase in Agulhas leakage is associated with a reduction of precipitation in the WRZ. At the same time, changes in the position and the strength of the westerlies and, related to that, changes in the SLP pattern, likely impact precipitation through other mechanisms not investigated here, such as changes in cut-off lows or ridging highs (Ivanciu et al., 2022b).

## 6   Discussion and conclusions

In this paper, we analysed past and future precipitation trends as well as the impact of the Agulhas Current System on precipitation over southern Africa in the regional atmospheric model CCLM. Three simulations were used: a simulation driven by an atmospheric reanalysis data set and two simulations driven by the global coupled climate model FOCI covering the past (1951-2013) and future (2014-2099).

Our analysis reveals the following:

- CCLM is capable of a good representation of the rainfall zones of southern Africa when comparing the hindcast simulation to observations and the driving reanalysis data set. Precipitation is underestimated over most of the domain.

Dosio et al. (2021b) found a good agreement of JRA-55 to other observational data sets for precipitation (1979-2018) and show generally comparable precipitation seasonal means in observational data sets for southern Africa. Gnitou et al. (2021) compared the annual cycle of a CCLM simulation to observations and a REMO simulation and found, like us, that CCLM is underestimating the monthly precipitation amounts. Nevertheless, CCLM performed better than REMO in terms of spatial added value coverage for all seasons (Gnitou et al., 2021). Munday and Washington (2018) found that CMIP5 models overestimate southern African rainfall (in austral summer) and underestimate rainfall over Madagascar due to the lower topography. This is contrary to the performance of our CCLM simulation. Panitz et al. (2014) found that the underestimation of CCLM (CORDEX-Africa) simulated rainfall peaks in the regions affected by the passage of the monsoon is a consequence of the wrong location of the monsoon centre, and underestimation of its intensity. This might explain the underestimation of rainfall in our CCLM simulation in the southeast (Madagascar and adjacent mainland) and the tropical western area of our domain.

– Precipitation trends in both the Summer Rainfall Zone and the Winter Rainfall Zone (SRZ and WRZ) were mostly positive in the past for the respective season but decreasing in some coastal areas of the SRZ, particularly at the southeast coast.

This agrees with the study of MacKellar et al. (2014) who analysed station data (1960-2010) and found an increase in rainfall over SRZ of South Africa for DJF and over WRZ in JJA, and with Lim Kam Sian et al. (2021) who found mainly wettening in South Africa (1901-2014) using CMIP6 simulations and gridded observations. Findings of Onyutha (2018) with CRU data agree with our results of mainly wettening, except for areas directly at the coast over the period 1901-2015 (annual precipitation) and more drying in coastal areas in austral summer. Analysis of station data for the WRZ showed a decrease for periods over the last 30 years (Roffe et al., 2021; Wolski et al., 2021). Also analysing precipitation trends over the last 30 years, Karypidou et al. (2022) found an underestimation of observed trends from CORDEX-Africa and CMIP5 models. Trends vary between CORDEX-Africa RCMs and the GCMs of CMIP5 and CMIP6 (Karypidou et al., 2022). Thus, trends over the past are spatially heterogeneous and strongly depend on the analysed time period as well as on the used data set. Even station data, gridded observations and reanalysis data sets provide varying precipitation trends to some degree.

– Future precipitation is projected to decrease over South Africa in both rainfall seasons. However, trends in the future are relatively weak which might be caused by the underestimation of precipitation amounts and trends by CCLM.

A decrease in future rainfall has also been found by e.g. Dosio et al. (2019), Rojas et al. (2019), Jury (2019), Seager et al. (2019), and Polade et al. (2017). Jury (2020) found that precipitation over the Agulhas Current is projected to decrease as well. CMIP6 models project an initial increase in the near future and then a decrease in western and eastern southern Africa (Lim Kam Sian et al., 2021). Global CMIP5 and CMIP6 models project a wetter future compared to regional models (CORDEX, CORDEX-CORE), with drying in western southern Africa and wettening in eastern southern Africa (Dosio et al., 2021a). Furthermore, Dosio et al. (2021a) found a decrease in precipitation frequency and an increase in

dry spells over southern Africa. In a previous study Dosio et al. (2019) analysed RCMs and got similar results and a good agreement of these RCMs and their driving GCMs.

– We applied a simple linear regression model to attribute the trends of SST and precipitation to the strength of the Agulhas Current and the strength of the Agulhas leakage. Our results show that the Agulhas Current System is linked to the SST in the southwest Indian Ocean and the South Atlantic and that it contributes to precipitation in South Africa. The reduction

in the strength of the Agulhas Current is linked to the reduction in precipitation along the southeast coast while the intensified Agulhas leakage is linked to the reduction in precipitation in the WRZ.

Cheng et al. (2018) linked the reduction in rainfall over southeast Africa in summer to the strength of the Agulhas leakage. Our results indicate an impact of the Agulhas Current System dominated by the strength of the Agulhas Current leading to the drying in that region. Furthermore, Cheng et al. (2018) found a linkage between the strength of the Agulhas

leakage to the meridional position and/or strength of the westerlies and the trade winds. This supports our results that the future drying of southern Africa is also linked to the strength of the Agulhas Current and the Agulhas leakage, which in turn are associated with the poleward shift and/or strengthening of the westerlies, especially under the currently most realistic SSP5-8.5 scenario.

– In addition to the Agulhas Current System as an oceanic driver, changes in the atmospheric circulation are leading to the

drying in the southern parts of southern Africa. Westerlies are projected to shift southward and strengthen, as previously found by e.g. Ivanciu et al. (2022a) and Tim et al. (2019). This displacement and intensification are accompanied by a poleward shift and intensification of the high-pressure systems of the oceans in austral winter. This implies a more southward passage of the frontal systems, responsible for rainfall in the WRZ, and thus can be linked to the drying in this region. In austral summer changes in the SLP are smaller but may cause less moisture transport from the ocean to

the southeast of our domain.

In summary, our simulations are suitable to analyse southern African precipitation, its changes and the impact of the Agulhas Current System. Coastal South African precipitation is projected to diminish over the 21st century and the strength of the Agulhas Current System is one of its drivers.

*Data availability.* CCLM and FOCI simulations are available upon request.

 **Appendix A: Bias of precipitation simulated by the CCLM hindcast**

Validation of the climatological mean precipitation of CCLM compared to GPCC and CRU and the bias relative to mean precipitation of CCLM with respect to GPCP, GPCC, and CRU.

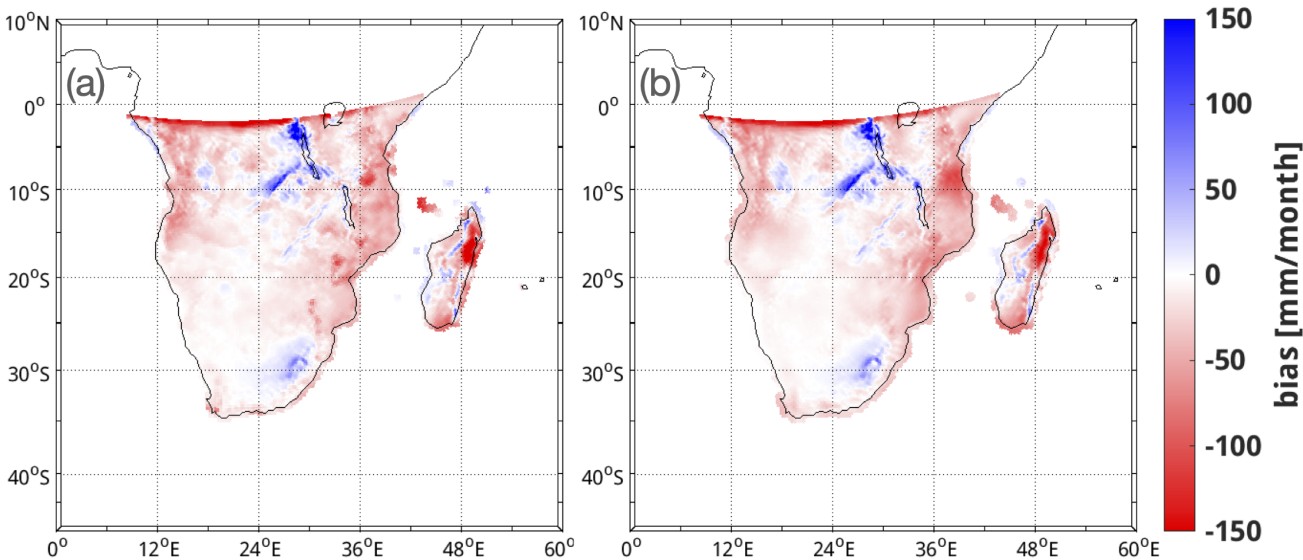

**Figure A1.** Model validation: model rainfall bias in [mm/month] given as the difference between modelled and observed rainfall from (a) CCLM and GPCC and (b) CCLM and CRU, respectively for the overlapping period 1958-04/2019.

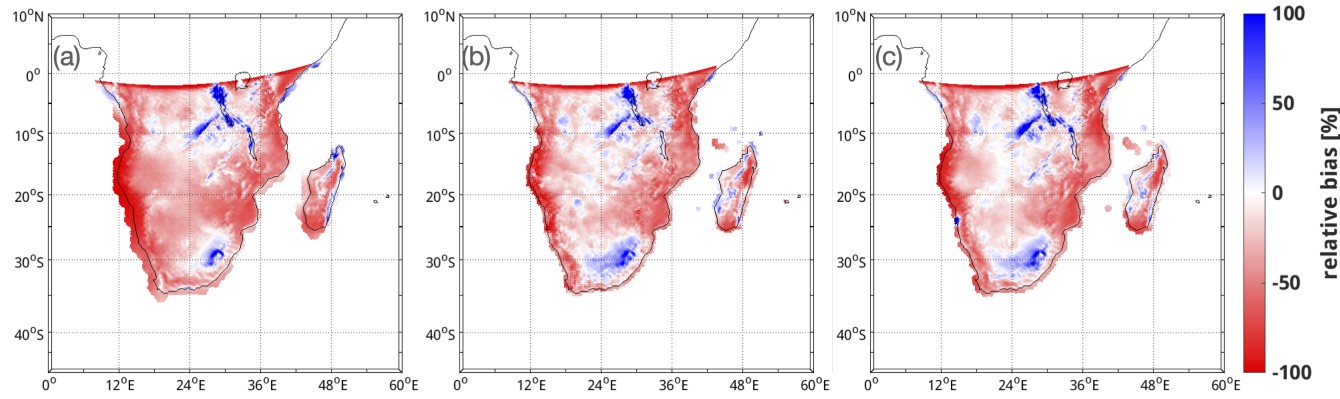

**Figure A2.** Relative bias in [%] of mean rainfall of CCLM versus (a) GPCP, (b) GPCC, and (c) CRU.

**Appendix B:  Attributing precipitation in the WRZ to the strength of the Agulhas Current and Agulhas leakage**

The impact of the Agulhas Current and the Agulhas leakage on precipitation in the WRZ looks very similar for austral winter
precipitation in the WRZ (Fig. B1) as for the whole year (Fig. 9). Precipitation decreases and Agulhas leakage and Agulhas
Current both impact this negative trend. The increase in Agulhas leakage strength contributes to the drying, as the precipitation
trend induced by the leakage is negative and the leakage is negatively correlated to precipitation. The Agulhas Current is also
negatively correlated with precipitation, so its negative trend would contribute to increased precipitation. However, as the total
precipitation trend is negative, Agulhas leakage should dominate here over the Agulhas Current. This is more pronounced here
than in figure 9 where the precipitation trend due to Agulhas Current is negative and correlations with the Agulhas Current are
positive in the southwesternmost part of the WRZ.

Regarding the precipitation trend over the WRZ in austral winter in the future and the impact of the Agulhas Current System
(Fig. B2), again, the relation looks similar to the analysis of the whole year (Fig. 10). Precipitation decreases and both Agulhas
leakage and Agulhas Current contribute to this decline. The Agulhas leakage is negatively correlated with precipitation, and as
its strength increases it causes a precipitation decrease. The Agulhas Current is positively correlated with precipitation and as
its strength weakens it causes a decrease in precipitation.

Thus, in the past, the impact of the Agulhas leakage seems to dominate over the Agulhas Current on winter rainfall. In the
future, both drivers contribute to the reduction of precipitation.

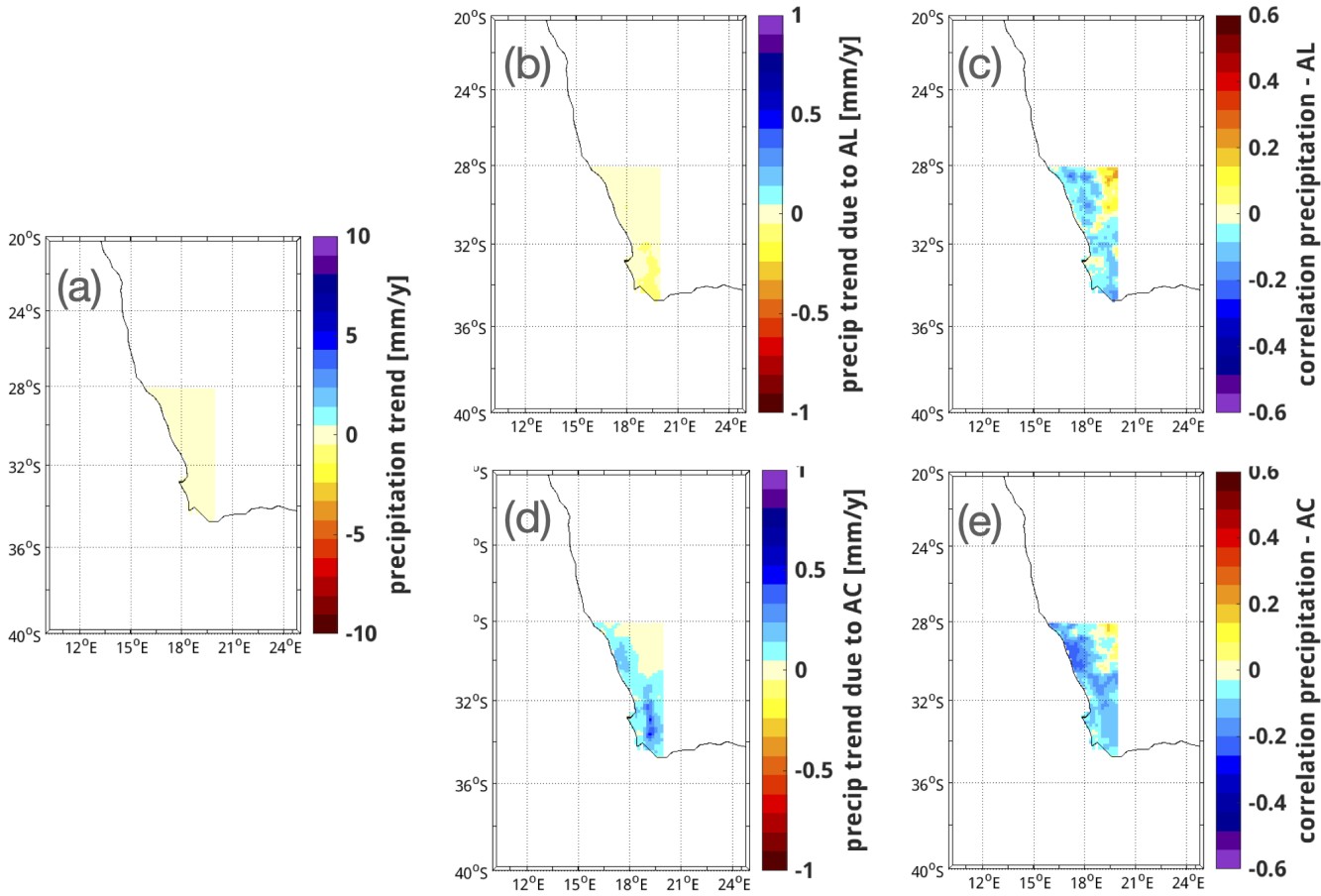

**Figure B1.** Impact of the Agulhas leakage and Agulhas Current on precipitation over the WRZ in austral winter in the historical simulation. (a) The trend in precipitation, (b) the portion of the precipitation trend attributed to the Agulhas leakage, (c) the correlation pattern of precipitation and the strength of the Agulhas leakage, (d) the portion of the precipitation trend attributed to the Agulhas Current, and (e) the correlation pattern of precipitation and the strength of the Agulhas Current. Colour bars of the correlation patterns (c, e) are reversed compared to the colour bars of the other subplots.

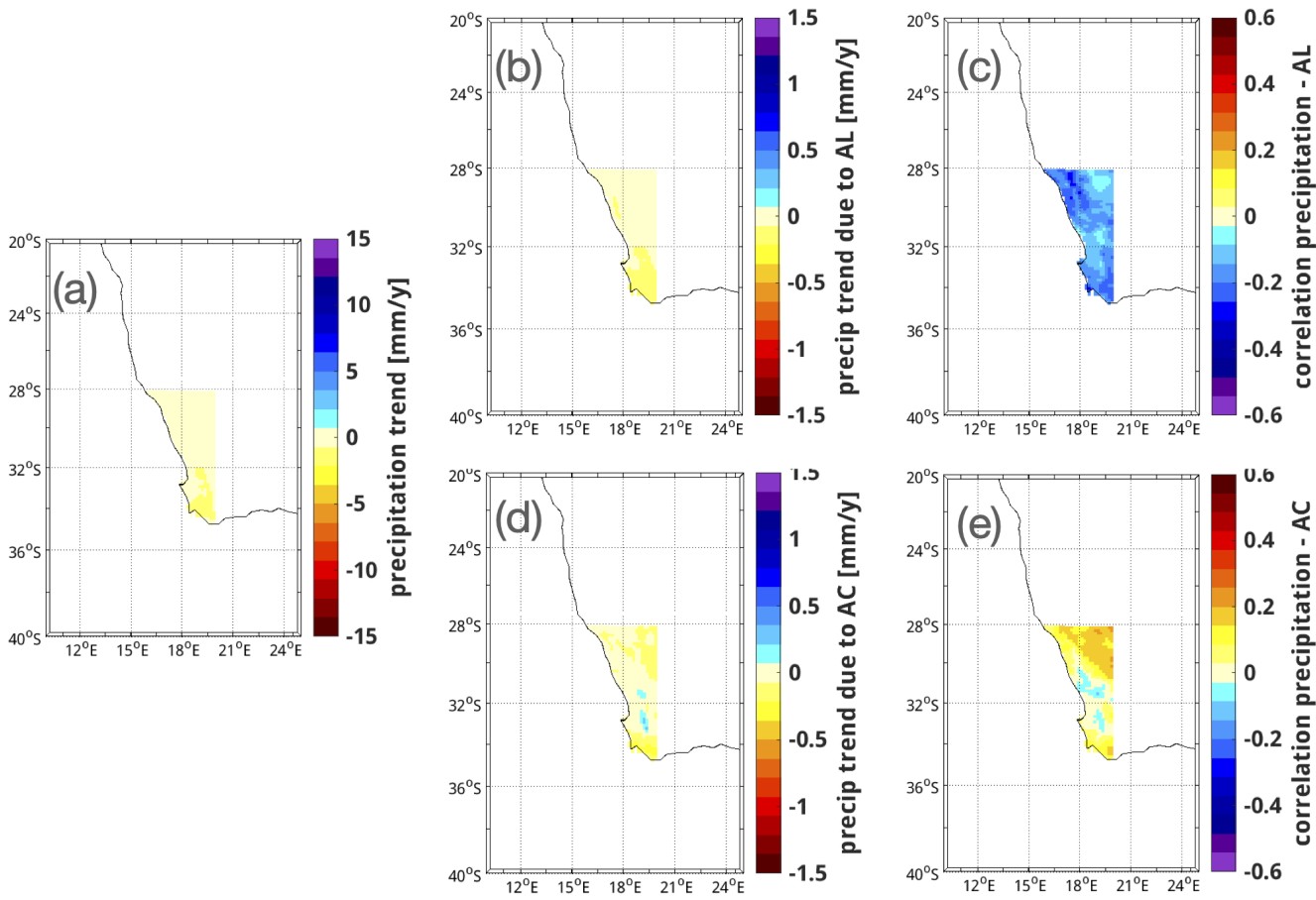

**Figure B2.** Impact of the Agulhas leakage and Agulhas Current on precipitation over the WRZ in austral winter in the scenario simulation. (a) The trend in precipitation, (b) the portion of the precipitation trend attributed to the Agulhas leakage, (c) the correlation pattern of precipitation and the strength of the Agulhas leakage, (d) the portion of the precipitation trend attributed to the Agulhas Current, and (e) the correlation pattern of precipitation and the strength of the Agulhas Current. Colour bars of the correlation patterns (c, e) are reversed compared to the colour bars of the other subplots.

*Author contributions.* NT, EZ, and BH designed the study and analysed the results, NT and EZ set up and ran the CCLM simulations. NT prepared the manuscript with contributions from EZ, BH and II.

*Competing interests.* The contact author has declared that neither she nor her co-authors have any competing interests.

*Acknowledgements.* We thank Sebastian Wagner and Beate Geyer for their support for the CCLM model setup. The CCLM model simulations have been performed at the German Climate Computing Center (Deutsches Klimarechenzentrum, DKRZ). The FOCI model simulations used in this study were performed with resources provided by the North-German Supercomputing Alliance (HLRN). The project received

funding from the German Federal Ministry of Education and Research (BMBF) of the SPACES-CASISAC project (grant 03F0796) and from the Helmholtz-Zentrum Hereon.

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
