# Peer review of "The impact of the Agulhas Current System on precipitation in southern Africa in regional climate simulations covering the recent past and future"

_Weather and Climate Dynamics, 2022_

## Community Comment (CC1)

**Open comment on wcd-2022-47:**
**Some concerns regarding the methodology and interpretation of a valuable study on Agulhas Current influence on South African precipitation.**

Stefaan Conradie

October 18, 2022

This is an important article on an understudied subject. I would strongly support the publication of a paper focussed on the evidence of influence of Agulhas Current changes on southern African precipitation from the high resolution model experiments described in this work. The comments provided herein are not intended as a complete evaluation of the submitted work. They focus primarily on claims made relating to my area of expertise. In particular, I have some significant concerns regarding the framing of the study, observational data used and representation and interpretation of some results, namely:

1. I would argue that the claims regarding recent WRZ rainfall trends made in the introduction (lines 31–33) are unclear, misleading and not well substantiated. Since this relates to relatively central conclusions of the study, a correction is required. More specifically:

   (a) Roffe et al. (2021) is cited to suggest a trend towards decreasing rainfall in recent decades over the WRZ using station data, whereas Roffe et al. (2021) do not conduct rainfall trend analysis at all, as far as I can tell. Many recent WRZ station rainfall trend assessments have been conducted, such as in Burls et al. (2019), Roffe et al. (2022) and Mahlalela et al. (2019).

   (b) MacKellar et al. (2014) is cited to support the assertion that "gridded observational data sets and reanalysis data indicate an increase ... [in] precipitation in the WRZ", as opposed to the decrease found by Wolski et al. (2020) with station data. However, MacKellar et al. (2014) also use station-based rainfall data and not, as far as I can tell, gridded observations. Furthermore, for the period 1960-2010 considered by MacKellar et al. (2014), significant rainfall trends are found at only one station in the WRZ; trends at all other stations are insignificant and small in magnitude. This is entirely consistent with 50-year SW cluster trends centred on 1985 depicted in Wolski et al. (2020, Fig 4).

   (c) My reading of the maps in Onyutha (2018) also indicate negative rainfall trends over the WRZ over 1990–2015.

   (d) Considering all studies I'm aware of that analyse recent rainfall trends over the WRZ (in addition to those previously listed: Ndebele et al., 2019; Du Plessis and Schloms, 2017; Hoffman et al., 2011; Kruger and Nxumalo, 2017; Otto et al., 2018; Seager et al., 2019; Pascale et al., 2020; Deitch et al., 2017; Conradie et al., 2022b; Zscheischler and Lehner, 2022), both station data and gridded gauge-based datasets indicate satellite-era drying, particularly since the 1990s. The primary disagreements between studies are because of either:

      i. differences in the time period or spatial domain considered (as noted in the conclusion); or
      ii. because the Climatic Research Unit (CRU) gridded rainfall dataset exhibits a growing negative rainfall bias after 2000, associated with a loss of data availability from wet reporting gauges Wolski et al. (2020).

(e) Hence, any statement about existing evidence on rainfall trends over the WRZ should rather focus on the general agreement of data sources regarding drying in recent decades, but perhaps note that the results are sensitive to spatial and temporal bounds used.

2. As noted by the first reviewer, the GPCP data (the version and a download link should be provided) used in this study covers too short a period to conduct robust trend analysis, especially given the well-established low-frequency variability in the WRZ, YRZ and SRZ (e.g. Dieppois et al., 2016). The short period of data coverage appears to be the result of the authors choosing to use daily rainfall data for these analyses, but it is not clear why daily data are required. Monthly GPCP data admittedly have a very coarse resolution, but the higher resolution daily data presented do not appear to accurately capture regions with large climatological rainfall gradients (see point 3 below). In addition, as far as I am aware, GPCP has not previously been used to study rainfall variability or change over the southern or western parts of southern Africa, where, as the authors note, the mechanisms driving rainfall occurrence and variability are different to the rest of southern Africa. Given that the authors pay considerable attention to rainfall changes over the WRZ during winter, validation should include comparison with a dataset that has previously been used over this region (and not CRU, given the biases pointed out previously), or whose regional veracity is demonstrated. GPCC data are available over a long period, at a much higher resolution and have been demonstrated to exhibit climatology, seasonality and variability characteristics deviating little from gauge-based observations (Wolski et al., 2020), so would seem an obvious choice. However, there may be reasons why other datasets are preferable in other parts of the domain, so the authors should of course use their discretion in selecting the most appropriate observations for their purposes.

3. The WRZ is mischaracterised in section 3, specifically in lines 126–129 and 138–140 and Figures 2 and 3. Specifically:

   - As far as I can tell, the bounds used for the WRZ domain in figures and analysis are not stated anywhere, nor is justification provided for the domain choice. It appears that an area that extends too far to the north-east has been selected, thus including places that receive very little rainfall, with all-year or late summer rainfall seasonality (cf. Roffe et al., 2019; Conradie et al., 2022a). This is reflected in the WRZ rainfall seasonality plots in Fig. 3, where the JRA-55 WRZ curve shows a secondary rainfall peak in March, suggestive of a late summer rainfall influence, whereas the GPCP data have an absolute minimum in September, suggestive of an interior YRZ seasonality. Furthermore, the amplitude of the seasonality in all datasets are much lower and the total annual rainfall much lower than most studies of the WRZ tend to find.

   - In the text the WRZ is equated to the "Western Cape region", despite the WRZ covering less than half the Western Cape provincial area and extending beyond its borders. Without a specfic definition, the "Western Cape region" is likely to be assumed by readers to correspond approximately to the South African province by that name. Furthermore, the WRZ is described as a "dryer region". Whereas some of the driest recording gauges in South Africa are located in the WRZ, the wettest recording gauge in the country is also situated in the WRZ and there are multiple gauges in the domain recording in excess of 1500mm per annum on average (Lynch, 2004; Slingsby et al., 2021). The South-Western Cape, as generally conceived, is certainly not a uniformly dry area, as appears to be suggested.

   - Furthermore, the relatively coarse resolution GPCP data do not adequately capture the regions of high (> 3mm/day from other datasets) rainfall in the south-west, the Drakensberg or eastern Madagascar (Fig. 2 and 3). CCLM appears to do better over the Maloti-Drakensberg of Lesotho than the reference data.

   - The fact that absolute scales are used in all panels of Fig. 2 and that the area on which most of the paper focuses on is dry relative to the Congo Basin and northern Madagascar means that the relative amplitude of the CCLM biases relative to GPCP are difficult to ascertain. In understanding trend and variability influence, relative bias is at least as important to assess as absolute biases.

- No mention appears to be made of the fact that the amplitude of the WRZ seasonal cycle is about 2x larger in the hindcast simulation than the historical simulation.

I therefore cannot concur with the authors that they demonstrate that "CCLM represents well the annual cycle of both rainfall regions SRZ and WRZ"; that "precipitation is generally realistically represented in the southern part of [their] domain"; or that "larger biases are mainly out of [their] focus region".

4. Whereas it is noted in the work that the YRZ occurs along the South Coast, it is not clear why in the analysis only the WRZ is distinguished from the rest of the domain, which appears to be collectively considered as the SRZ. In addition, very wet areas near the Equator apparently included in the SRZ have a very different precipitation seasonality to those areas south of roughly 15°S. Whereas the text notes that these areas are not the focus of the study, why are these regions then used as part of the aggregation domain in Fig. 3? Or if they aren't, what domain bounds are applied for Fig. 3?

5. The purpose of section 4 in the study is not clear. The questions posed at the beginning of the section are not clearly answered, nor are they clearly related to the overall aims of the study. Figures 4 and 5 compare trends over very different time periods, meaning that results are not really directly comparable. This is noted in the text, but the comparisons are interpreted nonetheless. In addition, these figures do not indicate statistical significance or compare the magnitude of trends to interannual variability or climatology for the seasons, making assessment of their practical significance difficult. The JRA-55 trends also seems suspiciously large over the centre of the domain; there isn't perhaps a scaling error here? The statements in the text regarding the significance of JJA WRZ rainfall trends are not clear and do not appear consistent with the figures; Figure 5(d) for the historical simulation strongly suggests negative trends.

6. Regarding section 5:

- How can SST trends be directly and separately attributed to Agulhas Current or Agulhas Leakage volumes using a simple univariate linear least-squares regression model when the two variables appear to co-vary significantly and apparently are both associated significantly with anthropogenic warming? Surely the method may merely be detecting changes or variability associated with a covariate? This is indirectly acknowledged in lines 249–250: "the SST trends in the North Benguela Upwelling System are probably not directly impacted by the Agulhas Current, but rather climate change impacts both the Agulhas Current and the SSTs". But it is not clear then how the conclusion can be drawn in line 251 that "the Agulhas Current and the Agulhas leakage contribute to the warming trend of SSTs adjacent to southern Africa." Perhaps the framing of this section needs to be looked at?

- Again the significance of correlations are not mentioned or displayed in most cases, making assessment of the robustness of results difficult.

- Furthermore, the statement in lines 247–248 that "the strong SST trend in the Cape basin is possibly more linked to the Agulhas leakage" needs to be substantiated.

- In Fig. 9, are annual totals being considered here? And if so, why, when previously seasonal precipitation had been assessed?

- Lines 272–273: "The change in the dependency of Agulhas leakage strength on precipitation from the past to the future requires further investigation." Surely this is the incorrect direction of implied dependence?

7. Parts of the discussion in the conclusion seem like they would be better suited to the introduction or results sections. The statement: "Dosio et al. (2021b) found a good agreement of JRA-55 to other observational data sets for precipitation (1979-2018) and show generally comparable precipitation seasonal means in observational data sets for southern Africa" should not be in the conclusion, but rather should inform assessment in section 3 of the implications of the comparisons conducted. At

the scale that results are reported and conclusions drawn in this work (e.g. the south-east coast of South Africa, the South African South Coast or the WRZ), the cited paper does not appear to provide support for this claim, however.

More minor concerns are:

1. The statement on line 1–2; "It defines the seasons and it directly impacts one of the principal sources of income, agriculture" should be substantiated with a citation. It is also not clear what is meant by rainfall "defin[ing] the seasons".

2. It should be clarified in lines 16–18 that the far south-west (the WRZ) is unique in that *most of the annual precipitation* here occurs during the winter season; the all-year or year-round rainfall zone (YRZ), as noted in line 130, also receives significant rainfall contributions during winter, as do some other parts of the east coast (see, for example, papers reviewed by Roffe et al. (2019)). Also, while frontal rainfall contributes $\approx 90\%$ of winter rainfall in the core of the WRZ (Burls et al., 2019), it is not accurate to imply this is the only source of rainfall in the WRZ (see, e.g., Abba Omar and Abiodun (2020)).

3. As noted by the first anonymous reviewer, Jury (2020) and Jury (2015) are relevant to the discussion of rainfall and the Agulhas Current.

4. The motivating paragraph, lines 41–43, makes claims that probably require further substantiation and clarification.

5. Line 65: "Therefore, there is probably a large-scale common mechanism related to the increased radiative forcing that is behind this large-scale spatial pattern of precipitation reduction, and which is independent of changes in the Agulhas Current System." This mechanism has been widely studied by, for example, Seager et al. (2019) and Polade et al. (2017).

6. Line 81: The "broader scope" of this study should be more precisely detailed here.

7. Lines 128-131: The division of South Africa into 8 rainfall zones by the South African Weather Service in 1972 (Rouault and Richard, 2004) is only one of a range of subdivisions that have been proposed. The sentence should be rephrased to indicate this.

8. Variables in equations should be denoted by a single character or symbol (with sub- and/or superscripts as necessary), not multi-character strings, to avoid ambiguity (e.g. is $AL(t)$ a single function or is it $A(t) \times L(t)$?), as is standard practise in all physical sciences (see, e.g., IUPAC guidelines linked to on the Copernicus manuscript preparation page). Where words are used in subscripts they should not be italicised.

9. The "wetting" referred to in line 163 appears to be focussed in the western interior, not west coast.

10. Line 169: "The hindcast simulation and JRA-55 agree that precipitation in the WRZ has increased in the past." The figure depicts this for the JJA season only; it probably does not apply in the annual total due to reductions in the shoulder seasons seen in most datasets.

11. Line 183: "is linked to" should perhaps be "may be linked to", unless the authors demonstrate the nature of the link.

12. Line 288–289: Citation suggested.

**References**

Abba Omar, S. and Abiodun, B. J.: Characteristics of cut-off lows during the 2015–2017 drought in the Western Cape, South Africa, Atmos. Res., 235, 104 772, doi:10.1016/j.atmosres.2019.104772, 2020.

Burls, N. J., Blamey, R. C., Cash, B. A., Swenson, E. T., al Fahad, A., Bopape, M.-J. M., Straus, D. M., and Reason, C. J. C.: The Cape Town "Day Zero" drought and Hadley cell expansion, npj Climate and Atmospheric Science, 2, 27, doi:10.1038/s41612-019-0084-6, 2019.

Conradie, W. S., Wolski, P., and Hewitson, B. C.: Spatial heterogeneity in rain-bearing winds, seasonality and rainfall variability in southern Africa's winter rainfall zone, Advances in Statistical Climatology, Meteorology and Oceanography, 8, 31–62, doi:10.5194/ascmo-8-31-2022, 2022a.

Conradie, W. S., Wolski, P., and Hewitson, B. C.: Spatial heterogeneity of 2015–2017 drought intensity in South Africa's winter rainfall zone, Advances in Statistical Climatology, Meteorology and Oceanography, 8, 63–81, doi:10.5194/ascmo-8-63-2022, 2022b.

Deitch, M., Sapundjieff, M., and Feirer, S.: Characterizing Precipitation Variability and Trends in the World's Mediterranean-Climate Areas, Water, 9, 259, doi:10.3390/w9040259, 2017.

Dieppois, B., Pohl, B., Rouault, M., New, M., Lawler, D., and Keenlyside, N. S.: Interannual to inter-decadal variability of winter and summer southern African rainfall, and their teleconnections, Journal of Geophysical Research: Atmospheres, 121, 6215–6239, doi:10.1016/j.egypro.2016.11.209, 2016.

Du Plessis, J. and Schloms, B.: An investigation into the evidence of seasonal rainfall pattern shifts in the Western Cape, South Africa, J. S. Afr. Inst. Civ. Eng, 59, 47–55, doi:10.17159/2309-8775/2017/v59n4a5, 2017.

Hoffman, M. T., Cramer, M. D., Gillson, L., and Wallace, M.: Pan evaporation and wind run decline in the Cape Floristic Region of South Africa (1974-2005): Implications for vegetation responses to climate change, Clim. Change, 109, 437–452, doi:10.1007/s10584-011-0030-z, 2011.

Jury, M. R.: Passive Suppression of South African Rainfall by the Agulhas Current, Earth Interactions, 19, 1–14, doi:10.1175/EI-D-15-0017.1, 2015.

Jury, M. R.: Marine climate change over the eastern Agulhas Bank of South Africa, Ocean Science, 16, 1529–1544, doi:10.5194/OS-16-1529-2020, 2020.

Kruger, A. C. and Nxumalo, M. P.: Historical rainfall trends in South Africa: 1921 – 2015, Water SA, 43, 285–297, doi:http://dx.doi.org/10.4314/wsa.v43i1.12, 2017.

Lynch, S. D.: Development of a raster database of annual, monthly and daily rainfall for Southern Africa: Report to the water research commission, Tech. Rep. WRC Report 1156/1/04, Water Research Commission, Pretoria, South Africa, 2004.

MacKellar, N., New, M., and Jack, C.: Observed and modelled trends in rainfall and temperature for South Africa: 1960-2010, S Afr J Sci, 110, doi:10.1590/sajs.2014/20130353, 2014.

Mahlalela, P. T., Blamey, R. C., and Reason, C. J. C.: Mechanisms behind early winter rainfall variability in the southwestern Cape, South Africa, Clim. Dyn., 53, 21–39, doi:10.1007/s00382-018-4571-y, 2019.

Ndebele, N. E., Grab, S., and Turasie, A.: Characterizing rainfall in the south-western Cape, South Africa: 1841–2016, Int. J. Climatol., p. joc.6314, doi:10.1002/joc.6314, 2019.

Onyutha, C.: Trends and variability in African long-term precipitation, Stochastic Environmental Research and Risk Assessment, 32, 2721–2739, doi:10.1007/S00477-018-1587-0/FIGURES/8, 2018.

Otto, F. E. L., Wolski, P., Lehner, F., Tebaldi, C., Van Oldenborgh, J., Hogesteeger, S., Singh, R., Holden, P., Fučkar, N. S., Odoulami, R. C., and New, M.: Environmental Research Letters Anthropogenic influence on the drivers of the Western Cape drought 2015-2017 Anthropogenic influence on the drivers of the Western Cape drought 2015-2017, Environ. Res. Lett, 13, 124 010, doi:10.1088/1748-9326/aae9f9, 2018.

Pascale, S., Kapnick, S. B., Delworth, T. L., and Cooke, W. F.: Increasing risk of another Cape Town "Day Zero" drought in the 21st century, Proc. Natl. Acad. Sci., p. 202009144, doi:10.1073/pnas.2009144117, 2020.

Polade, S. D., Gershunov, A., Cayan, D. R., Dettinger, M. D., and Pierce, D. W.: Precipitation in a warming world: Assessing projected hydro-climate changes in California and other Mediterranean climate regions, Sci. Rep., 7, 1–10, doi:10.1038/s41598-017-11285-y, 2017.

Roffe, S. J., Fitchett, J. M., and Curtis, C. J.: Classifying and mapping rainfall seasonality in South Africa: a review, S Afr Geogr J, 101, 158–174, doi:10.1080/03736245.2019.1573151, 2019.

Roffe, S. J., Fitchett, J. M., and Curtis, C. J.: Quantifying rainfall seasonality across South Africa on the basis of the relationship between rainfall and temperature, Climate Dynamics, 56, 2431–2450, doi:10.1007/s00382-020-05597-5, 2021.

Roffe, S. J., Steinkopf, J., and Fitchett, J. M.: South African winter rainfall zone shifts: A comparison of seasonality metrics for Cape Town from 1841–1899 and 1933–2020, Theoretical and Applied Climatology, pp. 1–19, doi:10.1007/s00704-021-03911-7, 2022.

Rouault, M. and Richard, Y.: Intensity and spatial extension of drought in South Africa at different time scales, Water SA, 29, 489–500, doi:10.4314/wsa.v29i4.5057, 2004.

Seager, R., Osborn, T. J., Kushnir, Y., Simpson, I. R., Nakamura, J., and Liu, H.: Climate variability and change of mediterranean-type climates, J. Clim., 32, 2887–2915, doi:10.1175/JCLI-D-18-0472.1, 2019.

Slingsby, J. A., Buys, A., Simmers, A. D. A., Prinsloo, E., Forsyth, G. G., Glenday, J., and Allsopp, N.: Jonkershoek: Africa's oldest catchment experiment - 80 years and counting, Hydrol. Process., doi:10.1002/hyp.14101, 2021.

Wolski, P., Conradie, S., Jack, C., and Tadross, M.: Spatio-temporal patterns of rainfall trends and the 2015–2017 drought over the winter rainfall region of South Africa, Int. J. Climatol., p. joc.6768, doi:10.1002/joc.6768, 2020.

Zscheischler, J. and Lehner, F.: Attributing Compound Events to Anthropogenic Climate Change, Bulletin of the American Meteorological Society, 103, E936–E953, doi:10.1175/BAMS-D-21-0116.1, 2022.

---

## Author Response (AR1)

Dear Prof. Dr. Fischer,

Thank you very much for handling the review process for our manuscript.

You can find below our responses to the reviewers' comments and the changes we applied in the revised version of the manuscript.

The original comments by the reviewers are written in italics.

**Reviewer #1:**

*It is missing some useful references:*
*2015, Passive suppression of South African rainfall by the Agulhas Current, Earth Interactions, 19, 1-14*
*2019, South Africa's future climate: trends and projections, in The Geography of South Africa, eds. J. Knight and C.M. Rogerson, Springer Nature, Switzerland, 305-313.*
*2020, Marine climate change over the eastern Agulhas Bank of South Africa, Ocean Science, 16, 1529-1544.*

We added the first suggested reference to the following sentence in line 23-25:

The Agulhas Current System impacts the precipitation in southern Africa (Imbol Nkwinkwa et al., 2021; Nkwinkwa Njouodo et al., 2018; Cheng et al., 2018; Jury, 2015; Reason, 2001).

and added this sentence in lines 27-28:

The sea surface temperature (SST) of the Agulhas Current also impacts South African precipitation indirectly via El Niño-Southern Oscillation (ENSO) (Jury, 2015)

We added the second suggested reference to the following sentence in lines 353-354:

A decrease in future rainfall has also been found by e.g. Dosio et al. (2019), Rojas et al. (2019), Jury (2019), Seager et al. (2019), and Polade et al. (2017).

We added the third suggested reference in lines 354-355:

Jury (2020) found that precipitation over the Agulhas Current is simulated to
355 decrease too.

*in Fig 4a, 5a the observed GPCP rain trend is given 1997-2018, this period is too short. With trends over the sea masked, why not use the Chirps2 or ERA5 rainfall trend? these cover a longer time period*

As CHIRPS covers a relative short period too (1981 onwards), we added three more data sets to the validation analysis: ERA5 (as suggested by the reviewer) and additionally GPCC (available 1891 onwards) and CRU (available 1901-2020). We added a short description of these data sets in section 2 (lines 96-108):

[revised manuscript text omitted]

*in Fig 7a the (Hadley) observed SST trend should be compared with model trend*

We add the observed trend to the figure and compared it to the simulated SST trend (lines 247-250):
Compared to the observed SST trend (Fig. 7b), the simulated trend does not show a cooling in the Benguela Upwelling System either in west of the retroflection or between the Agulhas Current and the Return Current south of it. Nevertheless, both data sets, HadISST1 and CCLM, show a warming of the Agulhas Current and in the area of retroflection.

*in Fig 8a there is a SST warming in the Angola Dome in the tropical E. Atlantic which is said to be related to the Agulhas leakage, however this zone is where most models fail to correctly simulate the shift of anticyclonic winds and tropical rainfall. Thus changes in SST due to Agulhas leakage could be linked to a poleward shift in the subtropical ridge? or*

*inability of model to reflect the teleconnections? A useful reference on this subject: 2013, Climate trends in southern Africa, S. Afr. J. Science, 109, 53-63 - although using CMIP3 their Fig 1 shows the model bias that continues (with lesser values) in CMIP6.*

In our interpretation, the reviewer is suggesting that the deficiency of coarse resolution global models to replicate the shift of the anticyclone and tropical rain band could be due to the unrepresented effect of the Agulhas leakage in those models. This is an interesting hypothesis. We have investigated a bit further to what extent the variations of the Agulhas leakage impacts rainfall in the Eastern Tropical Atlantic. Usually, the bias of coarse resolution global climate models in the EBUS, in particular SSTs, has been attributed to an incorrect cloud parametrization or to a poorly resolved upwelling dynamics (see e.g doi: 10.1002/wcc.338), but the role of the Agulhas system is an interesting alternative, at least in the Benguela EBUS. We discussed this in view of the suggested reference. Essentially, we now show the expansion to ocean precipitation of the analysis presented in Figure 9:

Precipitation trends in the tropical east Atlantic are negative (also in the Angola Dome but much weaker) and so is the contribution of the Agulhas leakage and of the Agulhas Current. Correlations are negative with the Agulhas leakage and positive with the Agulhas Current. Thus, the strengthening of the Agulhas leakage and the weakening of the Agulhas Current seem to be linked to this reduction in precipitation. Therefore, we conclude that the SST trend in the Angola Dome in figure 8 may really be related to the Agulhas leakage and that the CCLM model is able to realistically simulate tropical rainfall.

[Figure]

Figure: Impact of the Agulhas leakage and Agulhas Current on precipitation over southern Africa in the historical simulation. (a) Trend in precipitation, (b) the portion of the precipitation trend attributed to the Agulhas leakage, (c) correlation pattern of precipitation and the strength of the Agulhas leakage, (d) the portion of the precipitation trend attributed to the Agulhas Current, and (e) correlation pattern of precipitation and of the strength of the Agulhas Current. Colorbars of the correlation patterns (c, e) are reversed compared to the colobars of the other subplots.

*in Fig 9c,e, 10c,e the color bars are reversed, which may correctly be interpreted, but the caption needs to provide a note on this.*

We now indicate the reversed colourbars in the figure caption.
Colorbars of the correlation patterns (c, e) are reversed compared to the colorbars of the other subplots.

**Reviewer #2:**

*L133: generally, in regional climate models, precipitation is overestimated over the Drakensberg region. E.g: the weather research forecast model (WRF, see Koseki et al. 2018). How about the CCLM? Why does your model have more inland precipitation?*

CCLM also overestimates precipitation over the Drakensberg region and the large lakes in the northeast of our domain (Fig. 2c). In the rest of the domain, precipitation is underestimated.

*L320: you state that "trends over the past are spatially heterogeneous and strongly depend on the analysed time period as well as on the used data set." How does your chosen period and your data influence your analysis?*

The heterogeneity of precipitation trends over the past in different data sets may impact our analysis. We included more observational and reanalysis data sets to compare to CCLM. Nevertheless, most data sets agree upon a wetting over most parts of the SRZ. Precipitation trends in the future are more robust, with projected drying of the whole region.
For the section on the impact of the Agulhas Current System on precipitation, results are robust as they show a stationary relation between both variables. The strengthening of the Agulhas leakage leads to drying in the WRZ and the weakening of the Agulhas Current leads to a drying over the south and southeast coast of South Africa. The stronger trends of both, the Agulhas Current and the Agulhas leakage, in the future compared to the past, may contribute to a homogeneous drying of the region in the future. We added a figure of the trend of Agulhas Current and leakage in a revised version.

*There is a whole system for the formation of winter and summer precipitation (see the introduction of Imbol Nkwinkwa et al. 2021). What is the percentage of the contribution of the Agulhas leakage to winter precipitation?*

Our analysis of the contribution of the Agulhas Current System on precipitation is not done for each season separately. Nevertheless, precipitation in the WRZ is typically strongest in winter. Therefore, we assume that the contribution of the Agulhas leakage is of the same range for winter precipitation only as for the annual precipitation, namely 1/10. We analyzed the contribution for the winter season separately and included it in the appendix. The following sentence has been added at lines 303-305:
An analysis of impact of the Agulhas Current System on the winter rainfall (shown in the appendix), indicates that the decline in austral winter rainfall in the past is driven by the Agulhas leakage and in the future by both Agulhas leakage and Agulhas Current.

And this appendix:
Appendix B: Attributing precipitation in the WRZ to the strength of the Agulhas Current and Agulhas leakage
The impact of the Agulhas Current and the Agulhas leakage on precipitation in the WRZ looks very similar for austral winter precipitation in the WRZ (Fig. B1) as for the whole year (Fig. 9). Precipitation decreases and Agulhas leakage and Agulhas Current both impact this negative trend. The increase in Agulhas leakage strength contributes to the drying, as the precipitation trend induced by the leakage is negative and the leakage is negatively correlated to precipitation. The Agulhas Current is also negatively correlated with

precipitation, so that its negative trend would contribute to an increase in precipitation. However, as the total precipitation trend is negative, Agulhas leakage should dominate here over the Agulhas Current. This is more pronounced here than in figure 9 where the precipitation trend due to Agulhas Current is negative and correlations with the Agulhas Current are positive in the southwestern most part of the WRZ.

Regarding the precipitation trend over the WRZ in austral winter in the future and the impact of the Agulhas Current System (Fig. B2), again, the relation looks similar to the analysis of the whole year (Fig. 10). Precipitation decreases and both Agulhas leakage and Agulhas Current contribute to this decline. The Agulhas leakage is negatively correlated with precipitation, and as its strength increases it causes a precipitation decrease. The Agulhas Current is positively correlated with precipitation and as its strength weakens it causes a decrease of precipitation.

Thus, in the past, the impact of the Agulhas leakage seems to dominate over the Agulhas Current on winter rainfall. In the future, both drivers contribute to the reduction of precipitation.

*The south coast of South Africa receives precipitation all year long (Engelbrecht et al. 2015; Engelbrecht and Landman, 2016). Could you tell from your analysis whether this phenomenon is due to the Agulhas Current or the Agulhas leakage?*

Unfortunately, we cannot disentangle this from our analysis. Nevertheless, regarding future precipitation, in figure 10d, it looks as if rainfall trends in the YRZ are impacted by the Agulhas Current and not that much by the Agulhas leakage (Fig. 10b).

Minor corrections:

*Line 90: define CCLM because it is the first use.*

Changed as suggested

*Line 97: define FOCI.*

Definition added (lines 113-116):
The other two CCLM simulations are driven by the coupled climate model FOCI (Flexible Ocean and Climate Infrastructure, Matthes et al. (2020)) with interactive ozone chemistry, and high, mesoscale-resolving, ocean resolution around South Africa, ran by our project partners at the research centre GEOMAR (Germany).

*Line 107: explain or give a reference for the method f-test.*

Added an explanation in lines 124-126:
For the statistical significance of these trends a significance level of $p=0.05$ was adopted using the method f-test, a test for the null hypothesis that the variance of two normal populations is the same.

*Figure 2 is the annual mean or the climatology?*

It is the climatology. It is the mean over the whole overlapping period 1997-2018. We changed this plot in the revised version and included more observational and reanalysis data sets.

*Caption figure 3, replace reanaylsis by reanalysis*

Changed as suggested.

*L157 replace a extenuated by an extenuated*

Changed as suggested.

*L180 replace "decreasing slightly close the the South African coast" by "decreasing slightly close to the South African coast*

Changed as suggested.

*L197-L203: precise the figures you are referring to*

We added a figure of the trends of the Agulhas Current and Agulhas leakage strength.

*L227: "to" is missing*

Changed as suggested.

**Revisions addressing the community comment that have not been already addressed are included here:**

*1. I would argue that the claims regarding recent WRZ rainfall trends made in the introduction (lines 31–33) are unclear, misleading and not well substantiated. Since this relates to relatively central conclusions of the study, a correction is required.*

We rewrote this part of the introduction in line 34-45:

Precipitation trends in the WRZ based on observations depend on the time period and on whether annual means or the actual rainfall season is analysed. A drying is detected for the winter rainfall season over the recent past (1987-2016) (Roffe et al., 2021) whereas a wettening was found over longer periods (1960-2010) for the winter rainfall season alone and for annual precipitation (MacKellar et al., 2014). Wolski et al. (2021) found negative trends of annual means for the long period 1900-2017 and for the recent past 1981-2017, positive trends for the period 1933-2014, and mixed trends for the periods 1981-2014 and 1933-2017. This shows that trends over periods including the drought 2015-2017 are generally negative. Ndebele et al. (2020) analysed the observed rainfall of one station in Cape Town and found again that trends depend on the considered period. Rainfall increased over the periods 1841-1900 and 1930-1970 but decreased when the long period from 1900 to 2016 is analysed. Gridded observational data indicate mostly an increase in the winter rainfall season for several time periods (Onyutha, 2018). Precipitation in the SRZ in austral summer (December-January-February, DJF) has either increased or no trend has been detected. This happens in both types of datasets, station observations and gridded fields (Onyutha, 2018; Kruger and Nxumalo, 2017; MacKellar et al., 2014).

*3. The WRZ is mischaracterised in section 3, specifically in lines 126–129 and 138–140 and Figures 2 and 3.*

The WRZ region in this study covers the domain 15°E - 20°E and 28°S - 35°S. This WRZ spans the region west of Cape Agulhas (20°E and 35°S) up to the South-African - Namibian border at the west coast (as indicated in the blue line in Figure1 Roffe et al. (2021)). The choice of a rectangular domain lead to the inclusion of a small part of the YRZ and, yes, the WRZ includes regions of very little rainfall. The WRZ region is, of course, the same for all data sets. We added the geographic coordinates to the manuscript (lines 158-159):

The WRZ region in this study covers the domain 15 E - 20 E and 28 S - 35 S.

*In the text the WRZ is equated to the "Western Cape region", despite the WRZ covering less than half the Western Cape provincial area and extending beyond its borders. Without a specific definition, the "Western Cape region" is likely to be assumed by readers to correspond approximately to the South African province by that name. Furthermore, the WRZ is described as a "dryer region". Whereas some of the driest recording gauges in*

*South Africa are located in the WRZ, the wettest recording gauge in the country is also situated in the WRZ and there are multiple gauges in the domain recording in excess of 1500mm per annum on average (Lynch, 2004; Slingsby et al., 2021). The South-Western Cape, as generally conceived, is certainly not a uniformly dry area, as appears to be suggested.*

We changed the wording in the revised version from:

Dryer regions are the Namib and Kalahari deserts and the Western Cape region, the WRZ

to:

Dryer regions are the Namib and Kalahari deserts in the WRZ.

*Furthermore, the relatively coarse resolution GPCP data do not adequately capture the regions of high (> 3mm/day from other datasets) rainfall in the south-west, the Drakensberg or eastern Madagascar (Fig. 2 and 3). CCLM appears to do better over the Maloti-Drakensberg of Lesotho than the reference data. The fact that absolute scales are used in all panels of Fig. 2 and that the area on which most of the paper focuses on is dry relative to the Congo Basin and northern Madagascar means that the relative amplitude of the CCLM biases relative to GPCP are difficult to ascertain. In understanding trend and variability influence, relative bias is at least as important to assess as absolute biases.*

We added additional observational data sets not only for the trend validation but also for the bias of mean precipitation. We added the bias as percentages of the total precipitation in an appendix. Comparing to other observational data sets will improve the validation section. The following sentences regarding figure 2 were added (line 152-157):

Comparing CCLM to other observational data sets that cover a longer period but are available at monthly resolution, like GPCC and CRU, provides a very similar picture (see Fig. A1). The relative bias of CCLM to all three observational data sets (GPCP, GPCC, and CRU)(Fig. A2) is again of similar magnitude. But considering the relative bias reveals that deviations of simulated precipitation from observed ones are noticeable in the whole domain, also in our focus region, the southern part of the model domain and along the coasts, along which the Agulhas Current flows.

4. Whereas it is noted in the work that the YRZ occurs along the South Coast, it is not clear why in the analysis only the WRZ is distinguished from the rest of the domain, which appears to be collectively considered as the SRZ. In addition, very wet areas near the Equator apparently included in the SRZ have a very different precipitation seasonality to those areas south of roughly 15◦S. Whereas the text notes that these areas are not the focus of the study, why are these regions then used as part of the aggregation domain in Fig. 3? Or if they aren't, what domain bounds are applied for Fig. 3?

Figure 3 is applied over the whole domain. As the YRZ is the transition zone between WRZ and SRZ and represents a small region, it is not separated from the large SRZ. Trends are given here anyway as spatially resolved plots and as a spatially averaged timeseries. We added a sentence to address the differences in precipitation trends in the YRZ-part from the rest of the SRZ in the revised version (lines 162-164).

Both the WRZ and the SRZ in this study include parts of the YRZ (all-year rainfall zone). This relatively small transition zone between the WRZ and SRZ is not analysed separately here. This may contribute to the relative flat annual cycle of the WRZ.

*Furthermore, the statement in lines 247–248 that "the strong SST trend in the Cape basin is possibly more linked to the Agulhas leakage" needs to be substantiated.*

> This conclusion is plausible, as the climate change signal would be of larger spatial scale, but the reviewer is right that it is not completely substantiated by our results. We rewrote this sentence (lines 274-275).

Nevertheless, the link between the strong SST trend in the Cape basin and the Agulhas Current occurs possibly via the Agulhas leakage.

*Lines 272–273: "The change in the dependency of Agulhas leakage strength on precipitation from the past to the future requires further investigation." Surely this is the incorrect direction of implied dependence?*

> We of course meant the change of dependency of precipitation on the Agulhas leakage. We rephrased the sentence (299-300).

The change in the dependency of precipitation on Agulhas leakage strength from the past to the future requires further investigation.

*7. Parts of the discussion in the conclusion seem like they would be better suited to the introduction or results sections. The statement: "Dosio et al. (2021b) found a good agreement of JRA-55 to other observational data sets for precipitation (1979-2018) and show generally comparable precipitation seasonal means in observational data sets for southern Africa" should not be in the conclusion, but rather should inform assessment in section 3 of the implications of the comparisons conducted. At the scale that results are reported and conclusions drawn in this work (e.g. the south-east coast of South Africa, the South African South Coast or the WRZ), the cited paper does not appear to provide support for this claim, however.*

> Section 6 is titled Conclusions *and* Discussion, but we agree with the reviewer that this section needs a better structure, separating more clearly the discussion and the conclusion parts. We included a short Conclusions section without reference to previous results (lines 325-326, 338-339, and 351-352).

– CCLM is capable of a well representation of the rainfall zones of southern Africa when comparing the hindcast simulation to observations and the driving reanalysis. Precipitation is underestimated in most of the domain.
– Precipitation trends in both rainfall regions (SRZ and WRZ) are mostly positive in the past for the respective season but decreasing in some coastal areas of the SRZ, particularly the southeast coast.
– Future precipitation is projected to decrease for South Africa in both seasons. Trends in the future

More minor concerns are:

*2. It should be clarified in lines 16–18 that the far south-west (the WRZ) is unique in that most of the annual precipitation here occurs during the winter season; the all-year or year-round rainfall zone (YRZ), as noted in line 130, also receives significant rainfall contributions during winter, as do some other parts of the east coast (see, for example, papers reviewed by Roffe et al. (2019)). Also, while frontal rainfall contributes ≈ 90% of winter rainfall in the core of the WRZ (Burls et al., 2019), it is not accurate to imply this is the only source of rainfall in the WRZ (see, e.g., Abba Omar and Abiodun (2020)).*

> We clarified this in the revised version in lines 16-18.

Only the uttermost southwest of the continent receives most of its annual rainfall in the winter season (Chevalier and Chase, 2016). In this Winter Rainfall Zone (WRZ) rainfall is caused mainly by frontal systems moving with the westerlies (Reason, 2017).

*4. The motivating paragraph, lines 41–43, makes claims that probably require further substantiation and clarification.*

Changes in the winds, ocean currents and its link to anthropogenic change are the subject of the previous  paragraph. This is just a summarizing sentence. To be more clear, we merged it to that previous paragraph.

*5. Line 65: "Therefore, there is probably a large-scale common mechanism related to the increased radiative forcing that is behind this large-scale spatial pattern of precipitation reduction, and which is independent of changes in the Agulhas Current System." This mechanism has been widely studied by, for example, Seager et al. (2019) and Polade et al. (2017).*

We cited Seager et al. (2019) in lines 73-75. The mechanisms that these authors suggest for the Southern Hemisphere is essentially the expansion of the Hadley cell caused by increased atmospheric greenhouse gases. Surely, the Agulhas Current System is only one of the drivers of precipitation changes. Our results show that 1/10 of the drying can be attributed to the Agulhas Current System.

A common large-scale mechanism related to the increased radiative forcing and independent of changes in the Agulhas Current System is likely behind this large-scale spatial pattern of precipitation reduction. Previous studies have identified a diminished zonal moisture advection from the oceans located further west (Seager et al., 2019).

*7. Lines 128-131: The division of South Africa into 8 rainfall zones by the South African Weather Service in 1972 (Rouault and Richard, 2004) is only one of a range of subdivisions that have been proposed. The sentence should be rephrased to indicate this.*

We rephrased this sentence in lines 145-148:

South Africa can be divided into 8 rainfall zones: the North-Western Cape and the South-Western Cape constitute the WRZ, the South Coast, which has similar rainfall amounts during all months of the year (all-year rainfall region) and the SRZs Southern Interior, the Western Interior, the Central Interior, KwaZulu-Nata and the North-Eastern Interior (Rouault and Richard, 2003).

*11. Line 183: "is linked to" should perhaps be "may be linked to", unless the authors demonstrate the nature of the link.*

Changed as suggested.

*12. Line 288–289: Citation suggested.*

Reference has been added (Ivanciu et al., 2022b).

---

## Author Response (AR2)

Dear Prof. Dr. Fischer,

Thank you again for handling the review process for our manuscript. We are pleased to know that our revised version addressed all reviewer comments satisfyingly.

Here we list the changes we applied according to your comments:

*Line 233 "we look specifically at the impact of the strength of the Agulhas Current and Agulhas leakage on SST around". The end of the sentence ("around") is unclear to me. Consider rephrasing.*
> We rephrased the sentence to: „we look specifically at the impact of the strength of the Agulhas Current and Agulhas leakage on SST around southern Africa and precipitation over southern Africa."

*Line 258 suggest changing to "The Agulhas Current also contributes to about 1/6 of the SST trend"*
> Changed like suggested.

*Line 290 Given the high internal variability in trends of precipitation I suggest avoiding "will" for future regional precipitation trends. Suggestion "Thus, as the leakage will intensify (Ivanciu et al., 2022a) it contributes to a reduction of precipitation along the whole…."*
> Thus, as the leakage will intensify (Ivanciu et al., 2022a) it contributes to the reduction of precipitation along the whole…

*Line 325 CCLM is capable of a representing the rainfall zones of southern Africa well…*
> Changed to: CCLM is capable of a good representation of the rainfall zones of southern Africa…

*Line 382 Coastal South African precipitation is projected to reduce and the strength Agulhas Current System is one of its drivers.*
> Changed to: Coastal South African precipitation is projected to diminish over the 21st century and the strength of the Agulhas Current System is one of its drivers.

*I encourage the authors to have the manuscript proof read again with a focus on English language.*
> We have proofread the manuscript again regarding the English language.